# Analysis of zebrafish periderm enhancers facilitates identification of a regulatory variant near human *KRT8/18*

Huan Liu[1,2,3]*, Kaylia Duncan[4], Annika Helverson[2], Priyanka Kumari[2], Camille Mumm[2], Yao Xiao[1], Jenna Colavincenzo Carlson[5], Fabrice Darbellay[6], Axel Visel[6,7,8], Elizabeth Leslie[9], Patrick Breheny[10], Albert J Erives[11], Robert A Cornell[2,4]*

[1]State Key Laboratory Breeding Base of Basic Science of Stomatology (Hubei-MOST) and Key Laboratory for Oral Biomedicine of Ministry of Education (KLOBM), School and Hospital of Stomatology, Wuhan University, Wuhan, China; [2]Department of Anatomy and Cell Biology, University of Iowa, Iowa City, United States; [3]Department of Periodontology, School of Stomatology, Wuhan University, Wuhan, China; [4]Interdisciplinary Program in Molecular Medicine, University of Iowa, Iowa City, United States; [5]Department of Biostatistics, University of Pittsburgh, Pittsburgh, United States; [6]Environmental Genomics and Systems Biology Division, Lawrence Berkeley Laboratories, Berkeley, United States; [7]U.S. Department of Energy Joint Genome Institute, Lawrence Berkeley Laboratories, Berkeley, United States; [8]University of California, Merced, Merced, United States; [9]Department of Human Genetics, Emory University School of Medicine, Atlanta, Georgia; [10]Department of Biostatistics, University of Iowa, Iowa City, United States; [11]Department of Biology, University of Iowa, Iowa City, United States

**\*For correspondence:**
liu.huan@whu.edu.cn (HL);
robert-cornell@uiowa.edu (RAC)

**Competing interests:** The authors declare that no competing interests exist.

**Abstract** Genome-wide association studies for non-syndromic orofacial clefting (OFC) have identified single nucleotide polymorphisms (SNPs) at loci where the presumed risk-relevant gene is expressed in oral periderm. The functional subsets of such SNPs are difficult to predict because the sequence underpinnings of periderm enhancers are unknown. We applied ATAC-seq to models of human palate periderm, including zebrafish periderm, mouse embryonic palate epithelia, and a human oral epithelium cell line, and to complementary mesenchymal cell types. We identified sets of enhancers specific to the epithelial cells and trained gapped-kmer support-vector-machine classifiers on these sets. We used the classifiers to predict the effects of 14 OFC-associated SNPs at 12q13 near *KRT18*. All the classifiers picked the same SNP as having the strongest effect, but the significance was highest with the classifier trained on zebrafish periderm. Reporter and deletion analyses support this SNP as lying within a periderm enhancer regulating *KRT18/KRT8* expression.

## Introduction

Orofacial clefting (OFC), which can include cleft lip, cleft palate, or both, is among the most common structural birth defects. Concordance for non-syndromic cleft lip with or without cleft palate (NSCLP) is about 50% in monozygotic twins, suggesting a strong genetic contribution to the etiology of OFC (*Little and Bryan, 1986*; *Takahashi et al., 2018*). Multiple genome-wide association studies (GWAS) have advanced our understanding of this contribution as multiple independent GWAS, and meta-analyses of them, have identified more than 40 associated loci (*Saleem et al., 2019*). However, GWAS methods cannot distinguish SNPs that directly influence risk (i.e., functional SNPs) from those

merely in linkage disequilibrium with such SNPs (i.e., rider SNPs). Functional SNPs that lie in non-coding DNA may alter the activity of tissue-specific enhancers. Such functional SNPs can be identified through systematic reporter assays: functional SNPs will have allele-specific effects on the enhancer activity of the encompassing element, while rider SNPs may not have this quality (*Liu et al., 2017a*; *Lidral et al., 2015*). A bioinformatics-based approach to predicting functional SNPs requires training a machine-learning classifier on a set of enhancers of the type that the SNPs are expected to impact. The trained classifier can subsequently be used to score any element for its similarity to the training set, and also to evaluate the effect of a SNP on the score of the element containing it. The gapped-kmer support vector machine (gkmSVM) is a version of supervised machine learning based on the enrichment weights of all 10-mers in a training set (*Ghandi et al., 2014*; *Ghandi et al., 2016*). Using gkmSVM, a SNP's impact on the score of a 19-base-pair (bp) element encompassing it is called its deltaSVM: SNPs that convert strongly-weighted (i.e., enriched or depleted) 10mers into unweighted ones will have large deltaSVM values (*Lee et al., 2015*). Importantly, deltaSVM values were found to correlate reasonably well with the effects of SNPs on enhancer activity in reporter assays (*Lee et al., 2015*). Therefore, the deltaSVM value prioritizes disease-associated SNPs for their likelihood of being functional. A challenge for the investigator can be acquiring an appropriate set of enhancers upon which to train the classifier.

A meta-analysis of OFC several GWASs in several populations and a single OFC GWAS in Han Chinese identified lead SNPs adjacent to the *KRT8* and *KRT18* genes (*Leslie et al., 2017*; *Yu et al., 2017*). These genes are highly expressed in periderm, a simple squamous epithelium that comprises the most superficial layer of embryonic skin and oral epithelium, among other tissues (*Vaziri Sani et al., 2010*; *Dale et al., 1985*; *Moll et al., 1982*). We reasoned that functional SNP (or SNPs) at this locus may disrupt a periderm enhancer. Here we revisit the meta-analysis results and find an additional 13 SNPs in this locus that have at least a suggestive statistical association to OFC. To prioritize these SNPs for functional tests using the deltaSVM method it would be optimal to train a classifier on a set of enhancers that are active in human palate-shelf periderm. To our knowledge, no periderm cell line, from any species, exists. However, primary periderm cells are readily isolated from zebrafish embryos. Although the DNA sequences of tissue-specific enhancers are rarely overtly conserved between mammals and fish, enrichment for specific transcription factor binding sites can be a conserved feature (*Fisher et al., 2006a*; *Gorkin et al., 2012*; *Chen et al., 2018*). If the binding site features of zebrafish periderm enhancers are conserved with those of human periderm enhancers then a classifier trained on the former could be used to conduct a successful deltaSVM-based screen for SNPs that disrupt the latter.

Supporting the promise of this approach, the genetic pathways that underlie periderm development in mice and zebrafish are shared, in spite of the fact that the embryonic origins of periderm in the two species are distinct. In mouse embryos, the periderm develops at embryonic day 9 (E9), well after gastrulation is complete, from underlying ectoderm that expresses the basal-keratinocyte marker Tp63 (*Richardson et al., 2014*). In zebrafish embryos the periderm (initially called the enveloping layer [EVL]) becomes a distinct lineage at about 4 hr post fertilization (hpf), shortly before gastrulation, and is derived from superficial blastomeres that do not express *tp63* (*Kimmel et al., 1990*). EVL cells proliferate and cover the entire animal until at least 7 days post fertilization (dpf); by 30 dpf, the periderm is replaced by a periderm-like epithelium derived from basal keratinocytes (*Lee et al., 2014*). The gene regulatory networks that govern differentiation of the murine periderm and zebrafish EVL share a dependence on IRF6 (Irf6 in zebrafish) and CHUK (Ikk1 in zebrafish) (*Richardson et al., 2014*; *Fukazawa et al., 2010*; *Sabel et al., 2009Li et al., 2017*). Differentiation of the zebrafish EVL also depends on Grhl paralogs, Klf17, and simple-epithelium keratins (e.g., Cyt1, Krt4, Krt8, Krt18) (*de la Garza et al., 2013*; *Liu et al., 2016*; *Pei et al., 2007*; *Miles et al., 2017*). We predict that differentiation human periderm is similarly controlled as genes encoding orthologs of these proteins are implicated in risk for orofacial clefting (e.g., *GRHL3*, *KLF4*, and *KRT18*) (*Leslie et al., 2017*; *Liu et al., 2016*; *Leslie et al., 2016*; *Peyrard-Janvid et al., 2014*). These findings, and the relative ease of isolating zebrafish periderm through cell sorting (*de la Garza et al., 2013*), motivated us to study zebrafish periderm enhancers.

Here we used ATAC-seq to identify a of a set of zebrafish periderm enhancer candidates, then evaluated enriched transcription-factor binding sites, and conduct ed reporter assays with binding-site deletion analyses. Interestingly, the enriched binding sites indicate novel members of the periderm gene regulatory network. For comparison, we used the same methods to identify sets of

mouse palate epithelium enhancer candidates and human oral epithelium enhancer candidates. We trained classifiers on each set, and used them to prioritize OFC-associated SNPs near the *KRT18* gene. Finally, we subjected the top-candidate SNP to additional tests of the possibility that it is the functional SNP at this locus.

## Results

### Identification of periderm-specific enhancers throughout the zebrafish genome

In *Tg(krt4:gfp)*$^{gz7TG}$ transgenic embryos GFP was reported to be present exclusively in the most superficial layer of the embryo, called the enveloping layer (EVL) or periderm, after 8 hr post fertilization (hpf) (*Gong et al., 2002*). Because a separate transgenic line built from the same element shows reporter expression in both basal and superficial epidermal layers at 54 hpf (*O'Brien et al., 2012*), we sectioned *Tg(krt4:gfp)*$^{gz7TG}$ embryos at 11 hpf (4-somite stage) and confirmed that GFP was only present at high levels in the periderm (*Figure 1A*). We dissociated such embryos at 11 hpf, isolated GFP-positive and GFP-negative cells, and performed ATAC-seq on both populations (*Figure 1A*). We then mapped the small (<100 bp) ATAC-seq fragments, which are indicative of nucleosome free regions (NFRs), within the zebrafish genome (*Buenrostro et al., 2015*). The concordance of peaks called between two replicates of this experiment was strong (*Figure 1—figure supplement 1*). At the majority of NFRs, the density of mapped reads was comparable in GFP-positive and GFP-negative cells, but at about 5% of elements (i.e., 12865 and 13947 peaks, respectively; *Supplementary file 1a*), the normalized density of mapped reads was enriched in one or the other cell type (log$_2$(fold change)>0.5 or <−0.5, FDR < 0.01); we refer to these elements as GFP-positive NFRs (*Supplementary file 1b*) and GFP-negative NFRs (*Supplementary file 1c*), respectively (*Figure 1B*). Consistent with previous reports, overall ATAC-seq signal was high at transcription start sites (*Buenrostro et al., 2015*; *Quillien et al., 2017*; *Figure 1—figure supplement 2A*), but the majority of cell-type-specific NFRs were located in intergenic regions (*Figure 1—figure supplement 2B*). In both GFP-positive and GFP-negative NFRs, the average evolutionary conservation was higher within NFRs than in flanking DNA (*Figure 1—figure supplement 3*).

ATAC-seq identifies nucleosome free regions (NFRs), which include active enhancers, active and inactive promoters, and CTCF-bound regions, some of which are insulators (*Buenrostro et al., 2013*; *Iwafuchi-Doi et al., 2016*). We reasoned that the subset of NFRs that are active enhancers and promoters would be flanked by nucleosomes with histone H3 acetylated on lysine 27 (H3K27Ac), a mark of active chromatin (*Creyghton et al., 2010*). We used published data sets from whole embryos at 8 hpf or 24 hpf (*Bogdanovic et al., 2012*). Although periderm comprises a small fraction of the embryo, we found examples of elements with ATAC-seq signal virtually specific to GFP-positive cells that nonetheless overlapped or were flanked by peaks of H3K27Ac signal detected in whole embryos (*Figure 1C*).

We split the GFP-positive NFRs into clusters of high or low H3K27Ac density detected in whole-embryo lysates at 8 hpf and/or at 24 hpf (*Bogdanović et al., 2012*) (i.e., clusters 1 and 2, respectively) (*Figure 1B*). In the H3K27Ac$^{High}$ cluster, the average H3K27Ac ChIP-seq signal dipped in the center of the NFR, consistent with NFRs being flanked by nucleosomes bearing the H3K27Ac modification (*Figure 1D*). The average density of ATAC-seq reads did not differ between the H3K27Ac$^{High}$ (*Supplementary file 1d*) and H3K27Ac$^{Low}$ (*Supplementary file 1e*) clusters (p>0.05, Kolmogorov–Smirnov test) (*Figure 1D*), indicating that nucleosome depletion alone does not signify an active regulatory element. We employed the Genomic Regions Enrichment of Annotations Tool (GREAT) (v 3.0) (*McLean et al., 2010*) (assignment rule: two nearest genes within 100 kb) to identify the sets of genes associated with each cluster. The set of genes associated with GFP-positive NFRs was strongly enriched for the Gene Ontology (GO) terms in the zebrafish anatomy category including 'EVL' and 'periderm' (e.g., at 10–10.3 hpf, *Figure 1E*, *Figure 1—figure supplement 5A*). The significance of the association was much stronger for GFP-positive NFRs in the H3K27Ac$^{High}$ cluster than in the H3K27Ac$^{Low}$ cluster (*Figure 1E*). H3K27Ac is a dynamic mark of enhancers, however filtering on GFP-positive NFRs that are H3K27Ac-positive at 8 hpf, at 24 hpf, of at 8 hpf and/or at 24 hpf, or H3K4me1-positive (a more stable mark of enhancers) at 8 hpf, all yielded sets of elements whose associated genes were enriched for the same GO terms at very similar significance levels

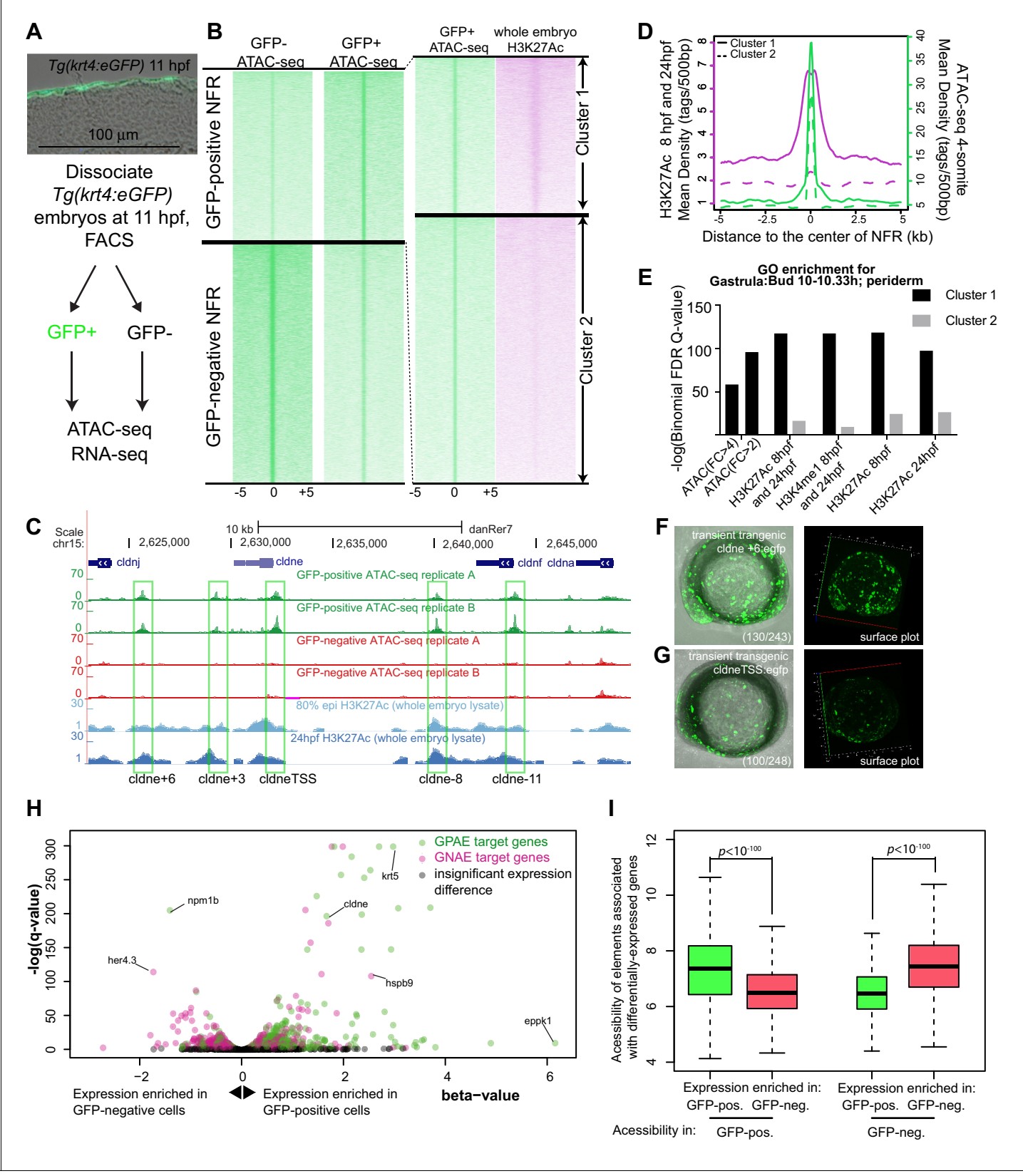

**Figure 1.** Identification of zebrafish GFP-positive active enhancers (zGPAEs) by integrating ATAC-seq and H3K27Ac ChIP-seq. (**A**) Transverse section of an 11 hpf (4-somite stage) *Tg(krt4:gfp)* embryo, showing GFP is confined to the superficial layer of cells, and workflow of ATAC-seq in periderm and *Figure 1 continued on next page*

Figure 1 continued

non-periderm cells. (B) Density plots of ATAC-seq results. Each line is centered on a nucleosome free region (NFR) with significantly more ATAC-seq reads in GFP-positive or GFP-negative cells; the majority of ATAC-seq peaks were not enriched in either cell type. Density plots also show H3K27Ac ChIP-seq signal in whole embryos at eight hpf and/or at 24 hpf data from *Bogdanovic et al. (2012)* at each of the GFP-positive NFRs; the latter are sorted in to those that overlap (or are flanked by within 100–1500 bp) peaks of H3K27Ac signal (cluster 1, 4301 elements) and or not (cluster 2, 7952 elements). (C) UCSC Genome browser tracks showing the ATAC-seq peaks in GFP-positive and GFP-negative cells, and H3K27Ac signal from whole embryos at eight hpf and at 24 hpf data from *Bogdanovic et al. (2012)* at the *cldne* locus. Boxes, examples of cluster one elements, also known as zebrafish GFP-positive active enhancers (zGPAEs). Elements are *cldne*+6 kb (zv9 : chr15:2625460–2625890), *cldne* +3 kb (chr15:2629012–2629544), *cldne* −8 kb (chr15:2639873–2640379), *cldne* −11 kb (chr15:2643578–2644160), and *cldne* TSS (chr15:2631981–2632513). (D) Plot of average density of H3K27Ac ChIP-seq signal (purple) and ATAC-seq signal (green). (E) GO enrichment for term 'Gastrula:Bud 10–10.33 hr; periderm' among NFRs enriched in GFP-positive cells with normalized fold change greater than 2 (ATAC(FC >2)) and 4 (ATAC(FC >4)), NFRs enriched in GFP-positive cells flanked or overlapped by 24hpf and 80% epiboly H3K27Ac ChIP-seq peaks (cluster 1) and or not (cluster 2), NFRs enriched in GFP-positve cells flanked or overlapped by 24hpf and 80% epiboly H3K4me1 ChIP-seq peaks (cluster 1) or not(cluster 2), NFRs enriched in GFP-positive cells flanked or overlapped by 24hpf H3K27Ac ChIP-seq peaks (cluster 1) or not (cluster 2), and NFRs enriched in GFP-positive cells flanked or overlapped by 80% epiboly H3K27Ac ChIP-seq peaks (cluster 1) or not (cluster 2). (F), (G) Lateral views of wild-type embryos at 11 hpf injected at the 1-cell stage with GFP reporter constructs built from (F) *cldne* +6 and (G) *cldne* transcription start site (TSS) elements. Left panels are stack views of the embryo, and right panels are surface plot for the embryos indicating most GFP signal is from the surface (periderm) of the embryos. Number in parentheses is the ratio of embryos with at least 10 GFP-positive periderm cells over injected embryos surviving at 11 hpf. (H) Volcano plot of RNA seq data, showing the expression of genes associated (by GREAT) with zGPAEs (green dots) or with zGNAEs (pink dots) in GFP-positive cells (beta-value >0) or in GFP-negative cells (beta-value <0). (I) Plot of accessibility scores of elements with differential accessibility (i.e., both zGPAEs and zGNAEs) associated with genes that are differentially expressed in GFP-positive and GFP-negative cells, showing that elements with increased accessibility in GFP-positive cells tend to be associated with genes whose expression is enriched in GFP-positive cells, and vice versa.

The online version of this article includes the following source data and figure supplement(s) for figure 1:

**Source data 1.** Density plot for ATAC-seq and H3K27Ac ChIP-seq, as plotted in *Figure 1D*.

**Source data 2.** Barchart for GO enrichment, as plotted in *Figure 1E*.

**Source data 3.** Scatter plot for the genes near GPAEs and GNAEs, as plotted in *Figure 1H*.

**Source data 4.** Box plot for the normalized chromatin accessibility of periderm- and non-periderm enriched genes in GFP positive or negative cells, as plotted in *Figure 1I*.

**Figure supplement 1.** Correlation of zebrafish periderm ATAC-seq two biological replicates.

**Figure supplement 2.** Annotation of ATAC-seq peaks relative to transcription start sites.

**Figure supplement 3.** Average Vertebrate PhastCons Score (danRer7 genome) at different distances from the center of nucleosome free regions (NFRs) in GFP-positive and GFP-negative (flow through) cells sorted from Tg(krt4:gfp) embryos at 11 hpf.

**Figure supplement 4.** Transient reporter assay validation for *cldne* +3, *cldne* −11, and *cldne* −8 elements.

**Figure supplement 5.** GO enrichment analysis for different clusters of GFP-positive or GFP-negative specific NFRs.

**Figure supplement 6.** Summary for RNA-seq for krt4:GFP-positive and krt4:GFP-negative cells at 4-somite stage.

**Figure supplement 7.** ATAC-seq near (A) keratin and (B) *her4* cluster genes.

(*Figure 1E*). By our definition GFP-positive NFRs have >2 fold more ATAC-seq reads in GFP-positive cells than in GFP-negative cells; perhaps surprisingly, the subset of GFP-positive NFRs with >4 fold more was less strongly associated with genes expressed in periderm than were GFP-positive NFRs overall (*Figure 1E*). In subsequent analyses, we used the set of GFP-positive NFRs positive for H3K27Ac signal detected in whole embryos at 8 hpf and/or at 24 hpf.

Although GFP-negative cells comprise a variety of non-periderm cell types, the set of genes associated with GFP-negative NFRs was also enriched for certain GO terms, including 'brain development' (*Figure 1—figure supplement 5B*).

We tested the enhancer activity of ten elements of the H3K27Ac^High cluster, each adjacent to a gene expressed in periderm, using reporter assays. For example, *cldne* is expressed in zebrafish periderm from 6 hpf to 15 hpf (www.zfin.org). We amplified five ~ 400 bp elements, located approximately +6 kb, +3 kb, −8 kb, −11 kb, and 0 kb from the transcription start site of *cldne* and engineered them into a reporter vector upstream of a minimal promoter and cDNA encoding GFP (*Fisher et al., 2006b*; *Figure 1C*). Embryos injected with these constructs exhibited mosaic GFP expression in the periderm at 11 hpf (e.g., *Figure 1F and G*, *Figure 1—figure supplement 4*). Four additional examples near other genes expressed in periderm (i.e., *gadd45ba*, *cavin2b*, *klf17*, and *ppl*) also had strong periderm enhancer activity (discussed later in the manuscript). Finally, in order to focus subsequent analyses on enhancers we filtered out elements that overlapped transcription start sites; the residual set of 3947 elements we refer to as zebrafish GFP-positive active enhancers

(zGPAEs) (*Supplementary file 1f*) . The analogous set of GFP-negative elements are GFP-negative active elements (zGNAEs).

To gain a genome-wide view of the association between zGPAEs and genes whose expression is enriched in periderm, we again sorted GFP-positive and GFP-negative cells from *Tg(krt4:gfp)* embryos at 11 hpf and generated expression profiles for both populations using RNA-seq. We identified 4331 genes with higher expression in GFP-positive cells and 4216 genes with higher expression in GFP-negative cells (q value < 0.05, beta <0, average TPM in GFP-positive cells > 1) (*Figure 1H*, *Figure 1—figure supplement 6A*). As expected, genes enriched in GFP-positive cells correlated positively with genes annotated at ZFIN, an online gene-expression atlas, as being expressed in the EVL (*Figure 1—figure supplement 6B*) (www.zfin.org). Differentially accessible elements associated with genes whose expression is enriched in GFP-positive cells had significantly higher average accessibility in GFP-positive cells than in GFP-negative cells, and vice versa (*Figure 1I*). For instance, there is a zGPAE near *cyt1*, a gene with higher expression in GFP-positive versus GFP-negative cells (*Figure 1—figure supplement 7A*) and there is a GNAE near *her4.3* with the opposite expression trend (*Figure 1—figure supplement 7B*). Some exceptions to the general pattern were observed (e.g., *hspb9* and *npm1b,* *Figure 1H*); this might reflect the fact that enhancers do not always regulate an adjacent gene. We conclude that most or perhaps all zGPAEs are enhancers active in periderm in embryos at 8 hpf to 24 hpf.

## Transcription factor binding sites overrepresented within zGPAEs

Using HOMER, we identified 12 short sequence motifs, corresponding to the preferred binding sites of specific transcription factors, that are enriched in zGPAEs and present in at least 5% them (*Figure 2A*); this list of zGPAE-signature motifs prompted testable hypothesis regarding the membership and structure of the periderm GRN. For instance, analysis of our RNA-seq profile of GFP-positive cells, and of available single-cell sequencing data (sc-seq) from zebrafish embryos (*Farrell et al., 2018*; *Wagner et al., 2018*), revealed transcription factors expressed in the EVL at 10–14 hpf; the subset of these that bind zGPAE-enriched motifs are candidates to participate in the periderm GRN (*Figure 2A*, 'best match'). In addition, clustering zGPAEs by virtue of the stage when the H3K27Ac signal is strongest (*Figure 2—figure supplement 1A*; *Bogdanović et al., 2012*) and then reassessing motif enrichment in each cluster revealed that the IRF6 site is more strongly enriched in early-acting zGPAEs (i.e., with H3K27Ac signal stronger earlier) (*Figure 2—figure supplement 1B*, cluster 1), and the TFAP2 and GATA sites in more strongly enriched in late-acting zGPAEs (*Figure 2—figure supplement 1B*, cluster 2 and 3), than in zGPAEs overall (*Figure 2A*). This implies that Irf6 acts earlier in the periderm GRN than do Tfap2 and Gata paralogs.

We predicted that zGPAE-enriched motifs would be essential for the periderm enhancer activity of zGPAEs. A zGPAE 3 kb downstream of the *cldne* transcriptional start site possesses a GRHL motif but lacks other signature motifs (*Figure 2B*). ATAC-seq data from GFP-positive cells included fewer TN5-mediated cleavage events within this motif than in flanking DNA, indicating that a transcription factor bound at the motif (*Figure 2C*; *Pique-Regi et al., 2011*). We amplified the zGPAE, deleted the motif by site-directed mutagenesis, engineered both the intact and motif-deleted versions into a GFP reporter vector (separately), and injected each into wild-type zebrafish embryos at the 1-cell stage. The periderm enhancer activity of the intact zGPAE was strong and specific at 11 hpf (4-somite stage) (*Figure 2D*), but that of the GRHL motif-deleted form was weaker (i.e., fewer injected embryos exhibited GFP in the periderm) (*Figure 2E,F*). Similarly, we identified zGPAEs in which motifs matching the KLF (*Figure 2—figure supplement 2A*), TFAP2 (*Figure 2—figure supplement 2B*) and C/EBP (*Figure 2—figure supplement 2C*) motifs were the only ones detected. In each case this sequence was protected from transposase access and its deletion reduced periderm enhancer reporter activity. Collectively, these assays support the assumption that zGPAE-enriched motifs are transcription factor binding sites essential for the function of periderm enhancers.

We then built a network using the best-match candidate transcription factors as nodes and linked the nodes using their putative target motifs as directional edges (from the regulating factor to its target motif, *Figure 2—figure supplement 3*). This network model raises additional testable hypotheses, including that GRHL transcription factors regulate expression of the other transcription factors enriched in periderm. Moreover, analyzing the genes associated with zGPAEs that contain specific motifs revealed that although only about 27% of zGPAEs contain a GRHL binding site, more than 70% of genes whose expression is enriched in GFP-positive cells are associated with a zGPAE

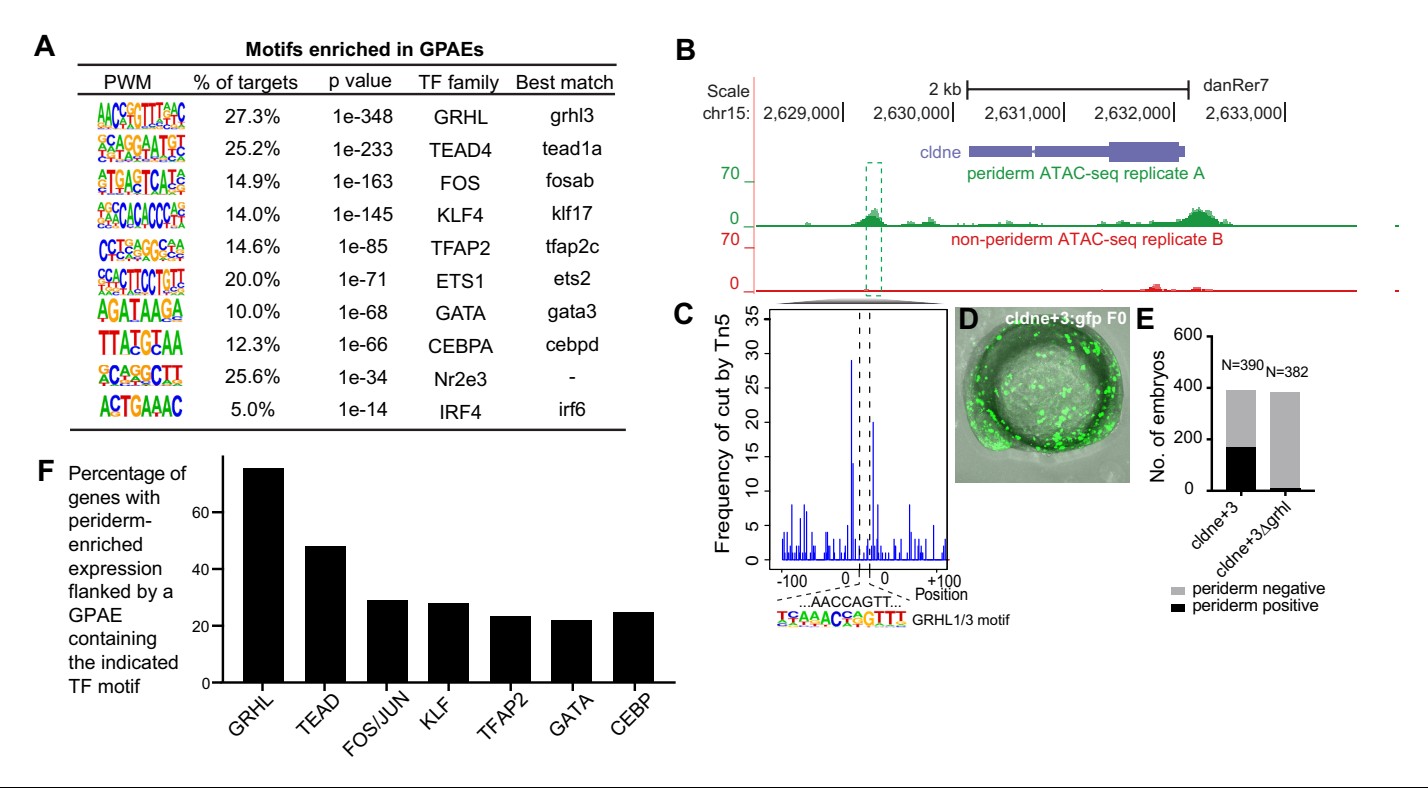

**Figure 2.** Features of zGPAEs. (**A**) Enriched motifs in zGPAEs. PWM, position weighted matrix. TF, transcription factors. Best match, transcription factor in the indicated family with highest expression in GFP-positive cells, whether or not the expression is enriched in GFP-positive cells in comparison to GFP-negative cells. (**B**) Genome browser view showing a GFP-positive nucleosome free region (NFR) about 3 kb downstream of the transcription start site of *cldne* gene. (**C**) Schematic of frequency of Tn5 cleavage sites at within this NFR, indicating reduced frequency of cleavage at a motif matching the GRHL binding site relative to in flanking DNA. (**D**) Confocal image of a wild-type embryo at 10 hpf (2-somite stage) injected at the one-cell stage with a reporter construct containing this NFR. (**E**) Bar chart showing number of embryos positive for GFP signal in the periderm after being injected with the intact reporter or one in which the GRHL motif was deleted. (**F**) Bar chart showing the percentage of genes whose expression is higher in GFP-positive cells than in GFP-negative cells that are flanked by a zGPAE possessing the indicated binding site.
The online version of this article includes the following figure supplement(s) for figure 2:

**Figure supplement 1.** Different clusters of H3K27Ac ChIP-seq at different developmental stages in zGPAEs.
**Figure supplement 2.** Transient reporter assay of (**A**) gadd45ba-3 with or without KLF motif, (**B**) cavin2b-+18 with or without TFAP2 motif and (**C**) klf17-+1.2 with or without C/EBP motif.
**Figure supplement 3.** Putative regulatory interactions of major periderm-enriched transcription factors governing transcriptomic state in periderm cells at 4-somite stage.
**Figure supplement 4.** Motif combination in GPAEs.

containing a GRHL site (*Figure 2F*). This implies that GRHL paralogs contribute to the regulation of most genes expressed in periderm.

To determine if particular combinations of signature motifs are over represented in zGPAEs we counted the numbers of zGPAEs with various two-motif or three-motif combinations (*Figure 2—figure supplement 4B and C*). While some combinations were present more frequently than others, genes associated with zGPAEs with the most frequent three-motif-combination and those associated with the least frequent three-motif combination both were highly enriched for with the GO term 'EVL (6–8 hr)'. Interestingly, however, the target genes of these two types of GPAEs rarely overlapped (*Figure 2—figure supplement 4D*). We also carried out unsupervised hierarchical clustering based on the frequency and combinations of motifs they contained (*Figure 2—figure supplement 4A*), but did not detect any striking patterns. Presumably distinct motif combinations reflect the deployment of transcription factors to effect expression of the associated genes in the EVL with specific timing, levels, or spatial constraint.

## A gapped k-mer support vector machine classifier trained on zebrafish periderm enhancer candidates

Towards comparing periderm enhancers in zebrafish and mammals, we sought to convert the pattern of enriched binding motifs into a scoring function. To this end we trained a supervised machine-learning classifier called gapped k-mer support vector machine (gkmSVM) on zGPAEs (*Ghandi et al., 2014*; *Figure 3A*). The trained classifier consists of a set of weights quantifying the contribution of each possible 10-mer to an element's membership in a training set. The resulting scoring function quantifies the degree to which a given test sequence resembles the training set (*Ghandi et al., 2014*). Five-fold cross validation on subsets of zGPAEs reserved from the training set revealed that the gkmSVM trained on zGPAEs had an area under the receiver operating characteristic curve (auROC) of 0.88, and area under the precision-recall curve (auPRC) of 0.87 (*Figure 3B*). The latter shows we can identify over 50% of all zebrafish enhancers (recall) at a false positive rate of under 10% (precision = 1-false positive rate). These performance measures compare favorably to those of sequence-based classifiers trained on tissue-specific enhancers in other studies (*Gorkin et al., 2012*; *Chen et al., 2018*) and support the validity of the parameters we chose to use for identifying zGPAEs.

The performance measures indicated that the classifier should be able to distinguish elements with periderm enhancer activity based on their sequence. To test this prediction, we partitioned the

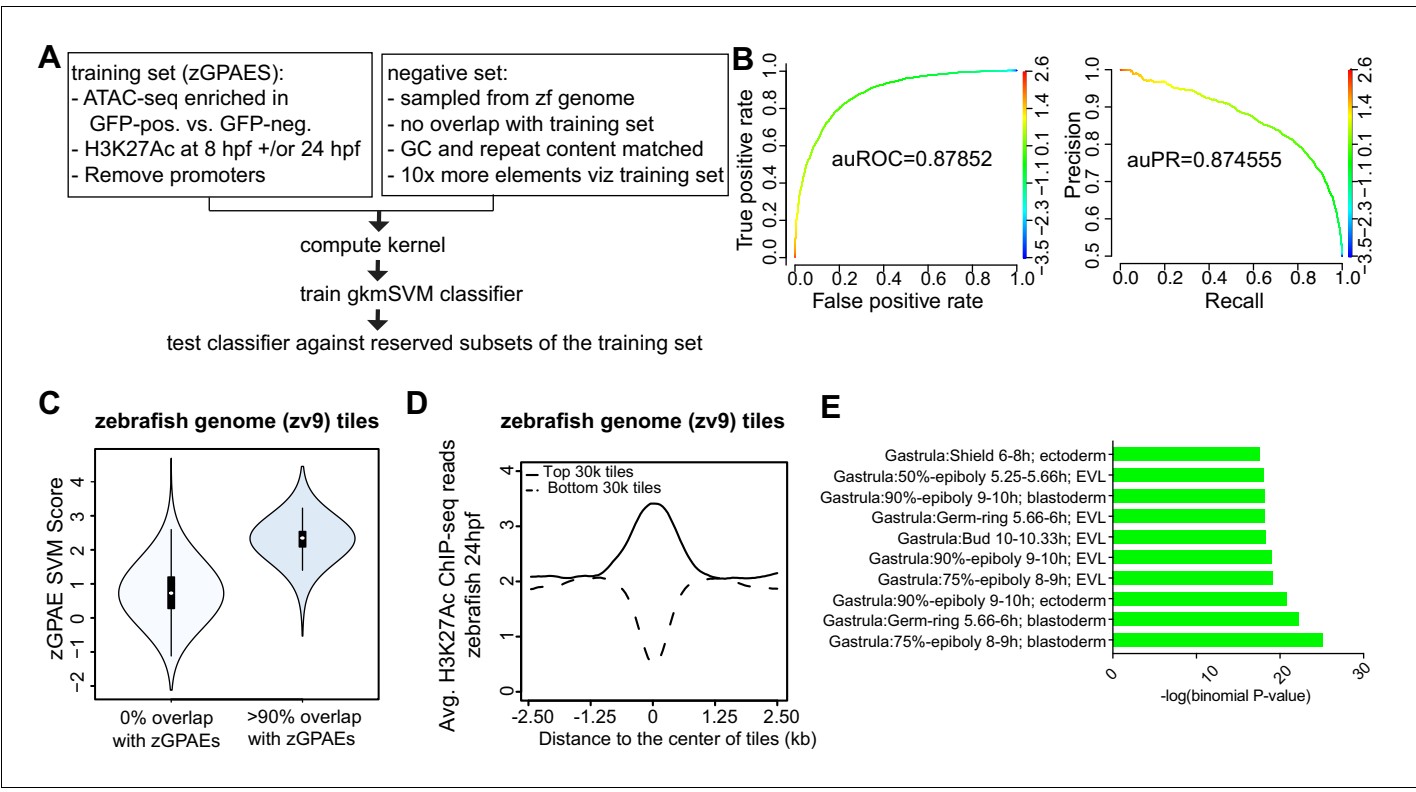

**Figure 3.** Training a gapped kmer support vector machine (gkmSVM) classifier trained on zGPAEs. (**A**) Pipeline for training and cross-validation of gkmSVM classifier on zebrafish periderm enhancer candidates. (**B**) Receiver Operating Characteristic (ROC) and Precision-Recall (PR) curves using the gkmSVM trained on zGPAEs. au, area under. Color of curves corresponds to SVM scores. (**C**) Violin plots showing SVM scores of zebrafish genome tiles with 0% or at least 90% overlapped with the training set (GPAEs). (**D**) Average H3K27Ac ChIP-seq reads at the 30,000 elements with the highest or lowest scores from the gkmSVM trained on zGPAEs. (**E**) GO enrichment assay for genes associated with the top-scoring tiles 10,000 tilesincluding those that overlap the training set.

The online version of this article includes the following source data and figure supplement(s) for figure 3:

**Source data 1.** Density plot for H3K27Ac ChIP-seq reads, as plotted in *Figure 3D*.
**Source data 2.** Barchart for GO enrichment assay, as plotted in *Figure 3E*.
**Figure supplement 1.** GO enrichment assay of gene expression for the top-scoring 10 K tiles that do not overlap zGPAEs.

genome into 400 bp tiles, each overlapping the preceding one by 300 bp, and scored each tile with the classifier. As expected, the average score of the tiles that overlap zGPAEs (i.e., the training set) was higher than that for tiles that do not overlap zGPAEs (*Figure 3C*). Moreover, the average H3K27Ac signal in 24 hpf embryos at the top scoring 30,000 tiles was higher than in the lowest-scoring 30,000 tiles (*Figure 3D*). Most importantly, genes associated with the top-scoring 10,000 tiles are enriched for the GO terms ectoderm, EVL, and periderm (*Figure 3E*), fulfilling our prediction. In addition, average expression of such genes was higher in GFP-positive versus GFP-negative cells in our RNA-seq profiles of these two cell types (p<1.46e-05, Mann-Whitney-Wilcoxon Test). Interestingly, the top-scoring 10,000 tiles that do not overlap zGPAEs are not enriched for GO terms related to EVL (*Figure 3—figure supplement 1*). Thus, even though the classifier overall has a low false discovery rate, given the large size of the genome there are many high-scoring elements are not periderm enhancers (false positives).

## Zebrafish periderm enhancers share a binding site code with mouse and human periderm enhancers

While tissue-specific enhancers are rarely conserved between mammals and zebrafish (with some exceptions *Visel et al., 2008*), inter-species reporter tests have shown that they nonetheless can be composed of the same binding site code (*Fisher et al., 2006a*). Therefore, we predicted that elements of the human genome that receive a high score from the classifier trained on zGPAEs will be enriched for human periderm enhancers. To test this notion, we divided the human genome into 400 bp tiles and scored each tile using the classifier trained on zGPAEs. We identified the top-scoring 0.1% bin of tiles (28,595 tiles) and examined their overlap with active enhancers, defined by ChIP-seq with antibodies to various chromatin-marks, in 125 cell/tissue types evaluated by the Roadmap Epigenomics project (*Kundaje et al., 2015*). Although periderm was not among the tissues evaluated by this project, top-scoring tiles were enriched within enhancers for several epithelial cell types more so than in other categories of enhancers (*Figure 4A*). For instance, the average H3K27Ac signal in the top 0.1% tiles was much higher in normal human epidermal keratinocytes (NHEK) than in a transformed lymphocyte cell line (GM12878) (*Figure 4B*).

If top-scoring tiles in the human genome include human periderm enhancers, then the set of genes associated with such tiles should be enriched for those expressed in periderm. While there is no available expression profile of human periderm, a recent single cell-seq analysis of a region in murine embryonic faces reported a cluster of 248 genes co-expressed with the canonical periderm marker *Krt17* (*Li et al., 2019*). Genes associated with the top-scoring 0.1% bin of tiles are enriched for those in this cluster (hereafter, mouse periderm) (hypergeometric p-value=0.044) (*Li et al., 2019*). Similarly, the zebrafish orthologs of such genes are expressed, on average, at higher levels in GFP-positive versus GFP-negative cells described above (Wilcoxon rank sum test, p-value=8.371e-06). Finally, we analyzed tiles of the mouse genome using the classifier trained on zGPAEs and found that tiles in the top-scoring 0.1% bin were strongly associated with genes expressed in mouse periderm (Fisher's Exact test, p=2.2e-16) (*Li et al., 2019*). These findings suggest periderm enhancers in zebrafish, human and mouse genomes are enriched for the same transcription factor binding sites.

An enhancer 9.7 kb upstream of the human *IRF6* transcription start site (i.e., *IRF6-9.7*) has been shown to drive reporter expression in oral periderm of transgenic mouse embryos (*Figure 4C*; *Fakhouri et al., 2012*). This element is of clinical interest as it harbors a mutation that co-segregates with Van der Woude syndrome in a family and that diminishes the periderm-enhancer activity of the element containing it (*Fakhouri et al., 2014*); it also harbors a single nucleotide polymorphism associated with risk for non-syndromic orofacial clefting (*Rahimov et al., 2008*). We engineered a 606 bp element within *IRF6-9.7* into the GFP reporter vector and injected it into wild-type embryos. In stable transgenic lines, GFP expression was detectable in the enveloping layer by shield stage (*Figure 4—figure supplement 1B–E*) and in periderm until at least five dpf (*Figure 4E*); it was also evident in pharyngeal epithelium at this stage (*Figure 4—figure supplement 1F and F'*). We created a double-transgenic embryo, harboring a tdTomato reporter whose expression is driven by the *krt4* promoter, and confirmed that tdTomato and GFP expression overlap (*Figure 4F*). Although *IRF6-9.7* is not overtly conserved to the zebrafish genome (*Figure 4C*), the part of it tested in mouse and zebrafish reporter assays contains a tile whose score matches the median score of tiles overlapping the training set (*Figure 3C*), and is in the top-scoring 1.0–1.5% bin of tiles in the human genome

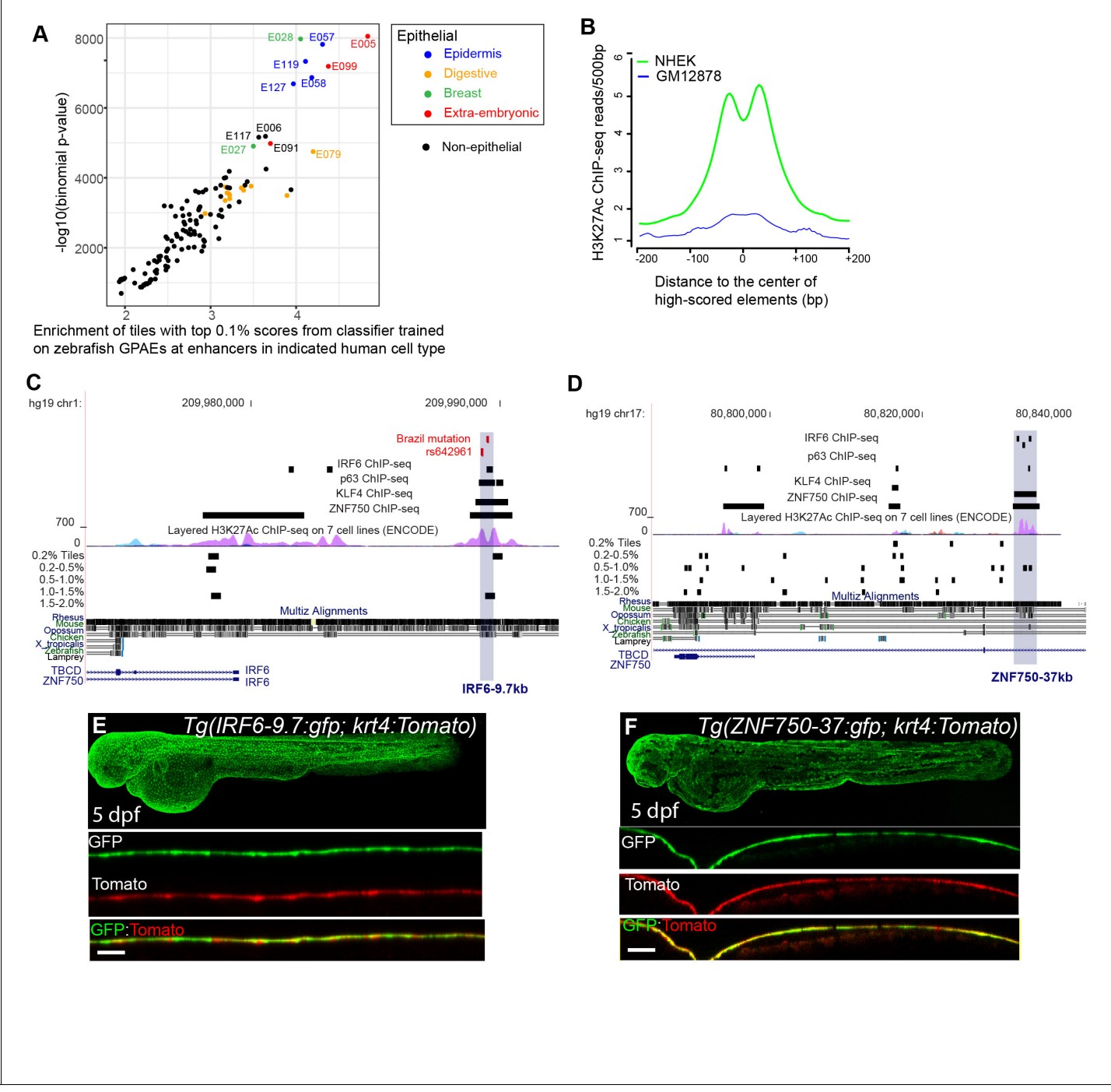

**Figure 4.** A classifier trained on zGPAEs applied to the human genome. (**A**) Enrichment of human genome tiles that receive a top 0.1% bin score using a zGPAE-trained classifier at enhancers active in the indicated cell type, as revealed by ChIP-seq to chromatin marks in the Roadmap Epigenomics project (*Visel et al., 2008*). -Such tiles are significantly enriched within a variety of epithelial enhancers. [E05: H1 BMP4 Derived Trophoblast Cultured Cells; E027: Breast Myoepithelial Primary Cells; E028: Breast variant Human Mammary Epithelial Cells; E057, E058: Foreskin Keratinocyte Primary Cells; E079: Esophagus; E091: Placenta; E099: Placenta amnion; E119, Mammary Epithelial Primary Cells (HMEC); E127:NHEK-Epidermal Keratinocyte Primary Cells]. (**B**) Average density of H3K27Ac ChIP-seq signal in NHEK and GM12878 cells (*Visel et al., 2008*) at top 0.1% tiles using a zGPAE-trained classifier. (**C**) Genome browser view focused on *IRF6-9.7*, also known as IRF6 multispecies conserved sequence 9.7 (MCS9.7) (hg19 chr1:209989050–209989824). A SNP within it, rs642961 (chr1: 209989270), is associated with risk for non-syndromic orofacial cleft. Brazil mutation refers to a rare mutation reported in a patient with Van der Woude syndrome (*Fakhouri et al., 2012*). This element has peaks of H3K27Ac, IRF6, TP63, and KLF4 ChIP-seq in normal human keratinocytes. Multiz Alignments of 100 vertebrate species shows it is conserved among mammals but not in zebrafishNonetheless it possesses tiles in the top 1.0-1.5% and 0.2% bins using a zGPAE-trained classifier (**D**) Genome browser view focused on

*Figure 4 continued on next page*

Figure 4 continued

ZNF750-37 (hg19 chr17:80832267–80835105). This element has similar ChIP-seq signature as *IRF6-9.7*, and like it is not overtly conserved to fish but possesses high-scoring tiles using the zGPAE-trained classifier. (**E**) GFP expression pattern of *Tg(IRF6-9.7:gfp; krt4:Tomato)* at five dpf. (**F**) GFP expression pattern of *Tg(ZNF750-37:gfp; krt4:Tomato)* at five dpf. Both of these human non-coding elements have periderm enhancer activity in zebrafish embryos.

The online version of this article includes the following source data and figure supplement(s) for figure 4:

**Source data 1.** Scatter plot for the enrichment of top scoring human genome tiles, as plotted in ***Figure 4A***.
**Source data 2.** Density plot for H3K27Ac ChIP-seq in NHEK and GM12878 cells within the top scoring human genome ties, as plotted in ***Figure 4B***.
**Figure supplement 1.** Detailed description of enhancer activity pattern of *Tg(IRF6-9.7:gfp)* and *Tg(ZNF750-37:gfp)*.
**Figure supplement 2.** Browser views of all loci with mcs9.7 ChIP-seq features.
**Figure supplement 3.** Reporter assay for human and zebrafish *PPL* elements predicted by zebrafish classifier.

(***Figure 4C***). Interestingly, the highest-scoring tile within the larger enhancer, marked by H3K27Ac in normal human epidermal keratinocytes (NHEK), is in the top-scoring 0.2% bin of tiles (***Figure 4C***).

We discovered a second periderm enhancer by searching the genome for elements sharing several ChIP-seq features of *IRF6*-9.7; specifically we filtered on strong H3K27Ac signal (ENCODE data) and peaks of IRF6 (***Botti et al., 2011***), KLF4 (***Boxer et al., 2014***), and TP63 (***Kouwenhoven et al., 2010***) binding, all assessed in normal human epidermal keratinocytes (NHEK). There are only five elements in the genome where all of these features converge, and each score in the top 2% bin or higher (***Figure 4—figure supplement 2***). We focused on one that lies 37 kb upstream of *ZNF750* (*ZNF750-37*) (***Figure 4D***). Chromatin configuration data indicate it binds to the *ZNF750* promoter in keratinocytes (***Rubin et al., 2017***). The mouse ortholog of *ZNF750* (i.e., *ZFP750*) is expressed in murine oral epithelium (***Richardson et al., 2017***), and periderm (***Li et al., 2019***), and the zebrafish ortholog *znf750* is expressed in EVL (***Farrell et al., 2018***). We amplified a 2.8 kb element overlapping the H3K27Ac signal from NHEK cells and made stable transgenic reporter fish. In them GFP expression is detectable in the enveloping layer starting at 5.25 hpf (50% epiboly) (***Figure 4—figure supplement 1I'***) and still visible at five dpf, although with lower intensity than in *Tg(IRF6-9.7:gfp)* transgenic animals at this stage (***Figure 4F***), consistent with lower expression levels of *znf750* in comparison to *irf6*. The highest scoring tiles in this apparent human periderm enhancer lies in the 0.5–1.0% bin (***Figure 4D***), supporting our prediction.

We found a third periderm enhancer 8.3 kb upstream of the transcriptional start site of *PPL* (encoding Periplakin) (*PPL-8.3*), whose mouse ortholog is highly expressed in periderm (***Li et al., 2019***) and contributes to epidermal barrier formation (***Sevilla et al., 2007***). Referring to ChIP seq experiments in NHEK cells, this element is bound by KLF4, ZNF750, and GRHL3, all transcription factors implicated in differentiation of keratinocytes (reviewed in ***Klein and Andersen, 2015***); it is on the flank of an island of H3K27Ac signal in NHEKs (ENCODE). In transient transgenic reporter assays in zebrafish it is a potent periderm enhancer (***Figure 4—figure supplement 3A and B***). We also amplified a zGPAEs 10 kb upstream of zebrafish *ppl* (*ppl-10*) (***Figure 4—figure supplement 3D and E***). Interestingly, deletion of KLF4 binding sites from the human element *PPL-8.3* or from the zebrafish element *ppl-10* strongly diminished the periderm enhancer activity in both cases (***Figure 4—figure supplement 3C and F***), suggesting that the two enhancers are at least partially functionally conserved. As predicted, the highest scoring tile within *PPL*-8.3 is in the top-scoring 1.5–2% bin.

To determine whether the shared binding sites reflect sequence homology between *ppl-10* and *PPL-8.3*, we performed sequence alignments. We found that a 467 bp core sequence from the zebrafish enhancer (plus-strand) is marginally more identical to a 400 bp core sequence from the human enhancer (plus-strand) relative to several control sequences including: the zebrafish minus-strand (reverse-complement), the non-biological reverse sequence, and non-biological sequences of similar lengths produced by Fisher-Yates shuffling of the plus-strand sequence (see Material and methods, ***Supplementary file 2a*** and ***Supplementary file 2b***). Furthermore, three-way alignments of the human and mouse plus-strand sequences with each of the zebrafish test and control sequences indicates that the zebrafish plus-strand engenders a need for a number of null characters (dashes) in the three-way alignments that is almost one standard deviation smaller than the controls (158 insertions versus an average number of insertions of 200.2 +/- 44.1 s.d. amongst minus-strand, reverse, and three Fisher-Yates shuffled sequences of the plus-strand). Last, the Hu_400+ and Zf_467+ pairwise-alignment has more 5 bp-long blocks of perfect identity (five such blocks) relative

to all five of the zebrafish controls (average two 5 bp-long blocs). Much of this potentially faint conservation overlies the elements conserved in the mammalian enhancer sequences (*Supplementary file 2b*). In summary, there is modest but detectable sequence homology between human *PPL-8.3* and zebrafish *ppl-10*.

## Defining sets of enhancers in mouse palate epithelium and a human oral epithelium cell line with ATAC-seq

Given the preceding findings, we reasoned that the gkmSVM classifier trained on the zebrafish periderm enhancers might be able to identify SNPs that disrupt periderm enhancers, and might perform as well as or better than classifiers trained on enhancers from other relevant tissues, such as mouse palate epithelium or a human oral epithelium cell line. To test this prediction, we dissected palate shelves from mouse embryos at embryonic day (E)14.5 and manually removed epithelium, both basal and periderm layers together, after brief incubation in trypsin (*Figure 5A*). Subsequently we subjected both the palate epithelium and the residual palate mesenchyme to ATAC-seq; there was a strong correlation among peaks in the three replicates (*Figure 5—figure supplement 1A*). In palate epithelium, we identified elements with more ATAC-seq reads than in palate mesenchyme (i.e., palate epithelium-specific NFRs, listed in *Supplementary file 1g and h*); these were binned as high- or low-density H3K27Ac signal (6079 and 8177 elements, respectively; listed in *Supplementary file 1i*) based on H3K27Ac ChIP-seq data from embryonic facial prominences (E14.5, ENCODE database, GEO:GSE82727) (*ENCODE Project Consortium, 2012*; *Figure 5B*). In the H3K27Ac$^{High}$ cluster, the average H3K27Ac ChIP-seq signal dipped in the center of the NFR (*Figure 5C*) as in zGPAEs. The average density of ATAC-seq reads was slightly higher in H3K27Ac$^{High}$ vs. H3K27Ac$^{Low}$ cluster (*Figure 5—figure supplement 1B*). Genes assigned to H3K27Ac$^{High}$ elements were strongly enriched for the GO term 'oral epithelium' ($\log_{10}$(binomial-FDR) <75), as were those assigned to H3K27Ac$^{Low}$ elements ($\log_{10}$(binomial-FDR) <49) (*Figure 5D*, *Figure 5—figure supplement 1C*). Significantly, we found H3K27Ac$^{High}$ cluster elements both near genes expressed at high levels in superficial palate epithelium (palate periderm) (e.g., *Krt17*, *Figure 5E*) and near those expressed in basal palate epithelium (e.g., *Krt14*, *Figure 5—figure supplement 1D*), showing that the isolated epithelium contained both of these layers. We also observed mesenchyme-specific NFRs in the H3K27Ac$^{High}$ cluster near genes whose expression is high in mesenchyme (e.g., *Runx2*, *Figure 5F*), and shared NFRs in the H3K27Ac$^{High}$ cluster near genes expressed in both (e.g., *Klf4*, *Figure 5—figure supplement 1E*). Elements in the H3K27Ac$^{High}$ cluster that do not overlap transcription start sites were named mouse palate epithelium active enhancers (mPEAEs) (*Supplementary file 1i*). HOMER revealed 18 short sequences for which mPEAEs are enriched and that are present in at least 5% of them (*Figure 5G*). Of note, six were predicted to be bound by transcription factors also predicted to bind one of the 11 zGPAE signature motifs (*Figure 5G*, bold text), suggesting epithelial enhancers in vertebrates share a set of core transcription factors.

Similarly, we carried out ATAC-seq, and H3K27Ac ChIP-seq on human immortalized oral epithelial cells (HIOEC) induced to differentiate by incubation in calcium. For comparison, we also carried out ATAC-seq on the human embryonic palate mesenchyme (HEPM) cell line . Focusing on ATAC-seq peaks that concorded among three replicates in each cell type, 31,296 NFRs present in HIOEC cells were absent in HEPM cells (*Supplementary file 1j*). Among such HIOEC-specific peaks, 15,972 overlapped (or were flanked by) H3K27Ac peaks called in two or more of the three replicates (cluster 1) (*Figure 5—figure supplement 2A*; listed in *Supplementary file 1k*) while 15,324 peaks neither overlapped nor were flanked (within 1500 bp) by H3K27Ac peaks (cluster 2). GO term enrichment revealed that genes associated with cluster 1 HIOEC-specific peaks were enriched for epithelial structure (*Figure 5—figure supplement 2B*), while cluster 2 did not exhibit such enrichment (*Figure 5—figure supplement 2C*). For instance, the *KRT17* gene is expressed in human epithelia, and chromatin regions within this locus were specifically open in HIOEC cells and overlapped with a human embryonic craniofacial super enhancer (*Figure 5—figure supplement 2D*). By contrast, chromatin regions within the *RUNX2* locus were specifically open in HEPM cells (*Figure 5—figure supplement 2E*). From cluster 1 we filtered out elements containing transcription start sites, and named the remaining subset human oral-epithelium active enhancers (hOEAEs). They were enriched for a set of binding sites, such as TEAD, JUN, C/EBP, GRHL and TFAP2; among these several were shared with zGPAEs and mPEAEs (*Figure 5—figure supplement 3*).

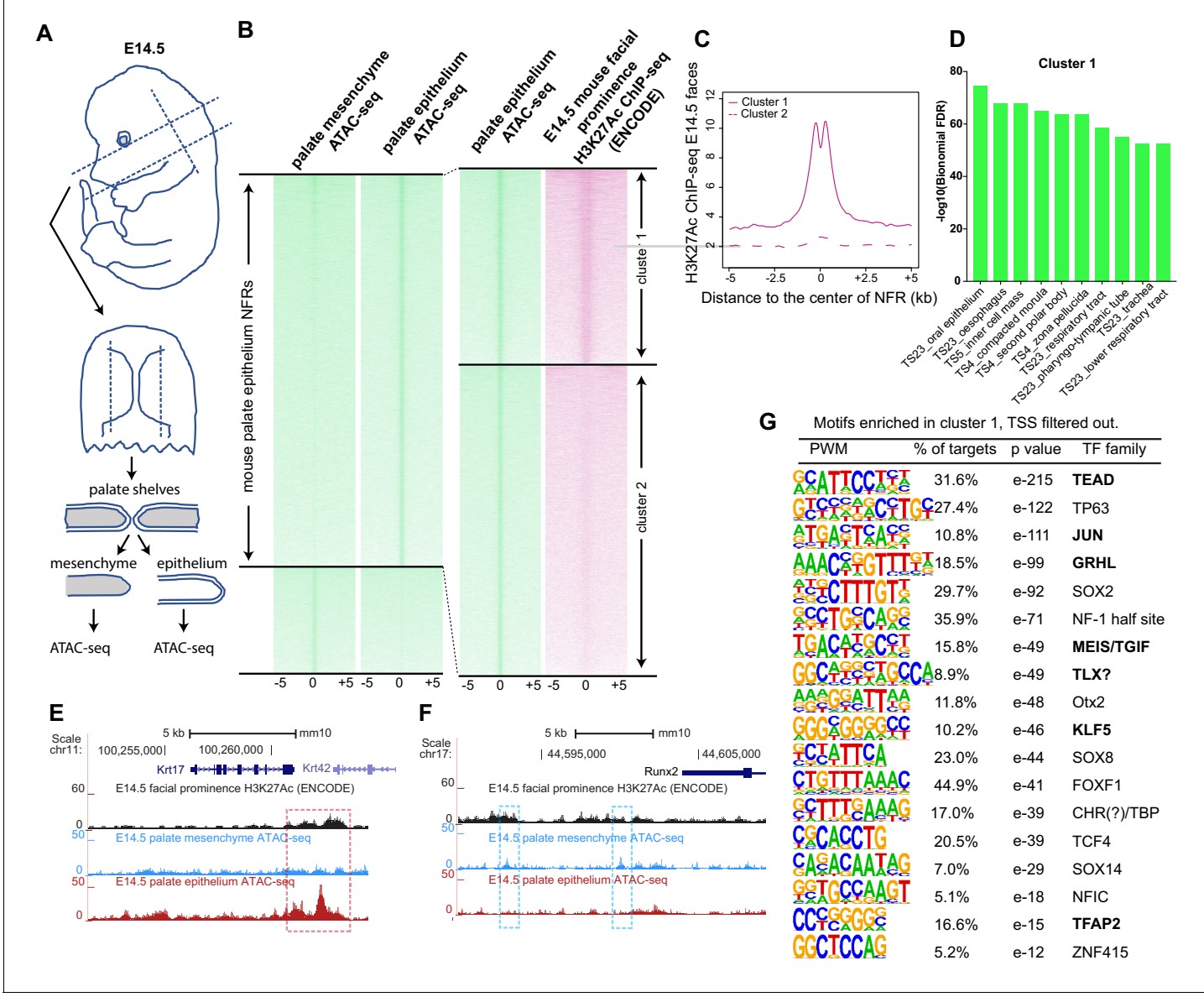

**Figure 5.** Identification of mouse embryonic palatal epithelium-specific active enhancers. (A) Workflow of ATAC-seq in epithelium and mesenchyme cells isolated from palate shelves dissected from E14.5 embryos. (B) Heatmap plots of ATAC-seq and E14.5 mouse facial prominence H3K27Ac ChIP-seq (*Klein and Andersen, 2015*) in tissue-specific NFRs. (C) Plot of average density of H3K27Ac ChIP-seq signal, showing higher signal at cluster 1 elements than cluster 2 elements. (D) GO enrichment (MGI mouse gene expression pattern) of genes associated with cluster 1 elements. (E and F) UCSC Genome browser views of the mouse genome (mm10 build) showing the ATAC-seq and H3K27Ac ChIP-seq signals near the *Krt17* and *Runx2* genes. Red box, an example of a mouse palate-epithelium active enhancer (mPEAE). Blue boxes, examples of mouse palate mesenchyme active enhancers (mPMAEs). (G) Motifs enriched in cluster 1 of E14.5 palate-epithelium specific NFRs with elements overlying transcription start sites removed (i.e., mPEAEs). Motifs shared with zGPAEs are in bold.

The online version of this article includes the following source data and figure supplement(s) for figure 5:

**Source data 1.** Density plot for H3K27Ac ChIP-seq in two clusters, as plotted in *Figure 5C*.
**Source data 2.** Barchart for GO enrichment, as plotted in *Figure 5D*.
**Figure supplement 1.** Concordance of replicates of mouse embryonic palatal epithelium ATAC-seq.
**Figure supplement 2.** Summary of ATAC-seq in HIOEC and HEPM cells.
**Figure supplement 3.** Motifs enriched in hOEAEs and shared among zGPAEs, mPEAEs and hOEAEs.

## Ranking OFC-associated SNPs using classifiers trained on zGPAEs, mPEAEs, and hOEAEs

Next, we used the classifiers trained on zGPAEs, mPEAEs, and hOEAEs to predict which single nucleotide polymorphisms (SNPs) associated with risk for orofacial cleft near the *KRT18* gene are most likely to disrupt an enhancer of the type upon which the classifier was trained. Revisiting our previously published GWAS data (*Leslie et al., 2017*), including imputed SNPs, we found 14 SNPs with at least suggestive p-values for association to risk for orofacial clefting (OFC) (p<1e-5) and in strong linkage disequilibrium with the lead SNP at this locus (i.e., SNPs 1–14) (SNP labels and p-values, *Supplementary file 1j*; *Figure 6A*). Functional SNPs are predicted to a) lie in enhancers active in a relevant tissue and b) have allele-specific effects on enhancer activity. To determine which SNPs lie in enhancers we evaluated published chromatin-state data from human embryonic faces (*Wilderman et al., 2018*) and 111 cell types characterized by the Roadmap Epigenomics project (*Kundaje et al., 2015*). Interestingly just three of the SNPs, i.e., SNP1, SNP2, and SNP13, lie in chromatin predicted to be active in one or more of these tissues while the others lay in relatively inert chromatin (*Figures 6B*, 9 representative Roadmap cell lines are shown). Using the classifier trained on zGPAEs, we calculated deltaSVM scores of the 14 SNPs, and for comparison, of 1000 additional SNPs within 100 kb (*Figure 6C and D*). The deltaSVM scores for most of the OFC-risk-associated SNPs were within one standard deviation of the median score of 1000 SNPs. By contrast, the deltaSVM score of SNP2 lower than that of 998 of 1000 randomly selected SNPs, and thus was an outlier (Bonferroni corrected p value = 0.028) (*Figure 6C and D*). Interestingly, SNP2 also had the strongest negative deltaSVM of all the OFC-risk-associated SNPs when the classifier was trained on mPEAEs or on hOEAEs, although in neither case were the deltaSVM values significant outliers in comparison to those of the 1000 randomly-selected SNPs (SNP2, Bonferroni corrected p values of 0.126 and 0.238, respectively) (*Supplementary file 3b-e*). Because *KRT18* is not expressed in palate mesenchyme, we used the classifier trained on mouse palate mesenchyme active elements (mPEAEs) as a negative control. As expected the deltaSVM for SNP2 was unremarkable (within the middle 50% of scores of 1000 SNPs) (*Figure 6D* and *Supplementary file 3b-e*). In conclusion, as DeltaSVM indicates that the risk-variant of SNP2 significantly lessens the periderm-enhancer score of the element containing it, SNP2 is the top-candidate for a functional SNP at this locus.

## Reporter assays in human oral epithelium cells support SNP2 being functional variant

Previously, upon training a gkmSVM classifier on melanocyte enhancers, the deltaSVM scores of SNPs within known melanocyte enhancers were found to correlate with the observed differences in reporter activity between t enhancer constructs tested in a melanocyte cell line (*Lee et al., 2015*). Therefore we predicted that, similarly, upon training a classifier on zGPAEs, mPEAEs, or hOEAEs, deltaSVM scores of SNPs within known epithelium enhancers would correlate with the differences in reporter activity between risk and non-risk constructs tested in a human basal oral keratinocyte cell line. Of the 14 OFC-risk associated SNPs at this locus, only SNP1 and SNP2 lie in chromatin marked as an active enhancer in normal human epidermal keratinocytes (NHEK) cells, which we predict are similar to oral keratinocytes (*Figure 6B*). SNP1 has a neutral deltaSVM when the classifier was trained on zGPAEs or mPEAEs (*Figure 6C and D* and *Figure 6—figure supplement 1*) and a deltaSVM of −3, in the lowest quartile of 1000 SNPs, when the trained on hOEAEs (*Figure 6D* and *Figure 6—figure supplement 1*). As mentioned above, SNP2 had a strongly negative deltaSVM with the classifier trained on zGPAEs, mPEAEs, and hOEAEs (*Figure 6C and D*). We amplified 700-base-pair elements centered on each SNP, engineered them to harbor either the risk-associated or non-risk associated allele of the SNP, introduced them (separately) into a luciferase-based reporter vector (with a basal SV40 promoter), and transfected these constructs into GMSM-K basal oral epithelium cells (*Gilchrist et al., 2000*). Both elements drove luciferase levels above background, suggesting that both SNPs lie in enhancers active in oral keratinocytes. SNP2 but not SNP1 had significant allele-specific effects on this enhancer activity, with the risk-associated variant driving lower reporter activity than the non-risk variant (*Figure 6E*). Thus, for these two SNPs, the deltaSVM scores and reporter effects were correlated, and SNP2 was further supported as being functional.

We next tested the prediction that the enhancer in which SNP2 lies regulates expression of *KRT8* and/or *KRT18*. We transfected GMSM-K cells with Cas9 ribonucleotide protein (RNP) and, in

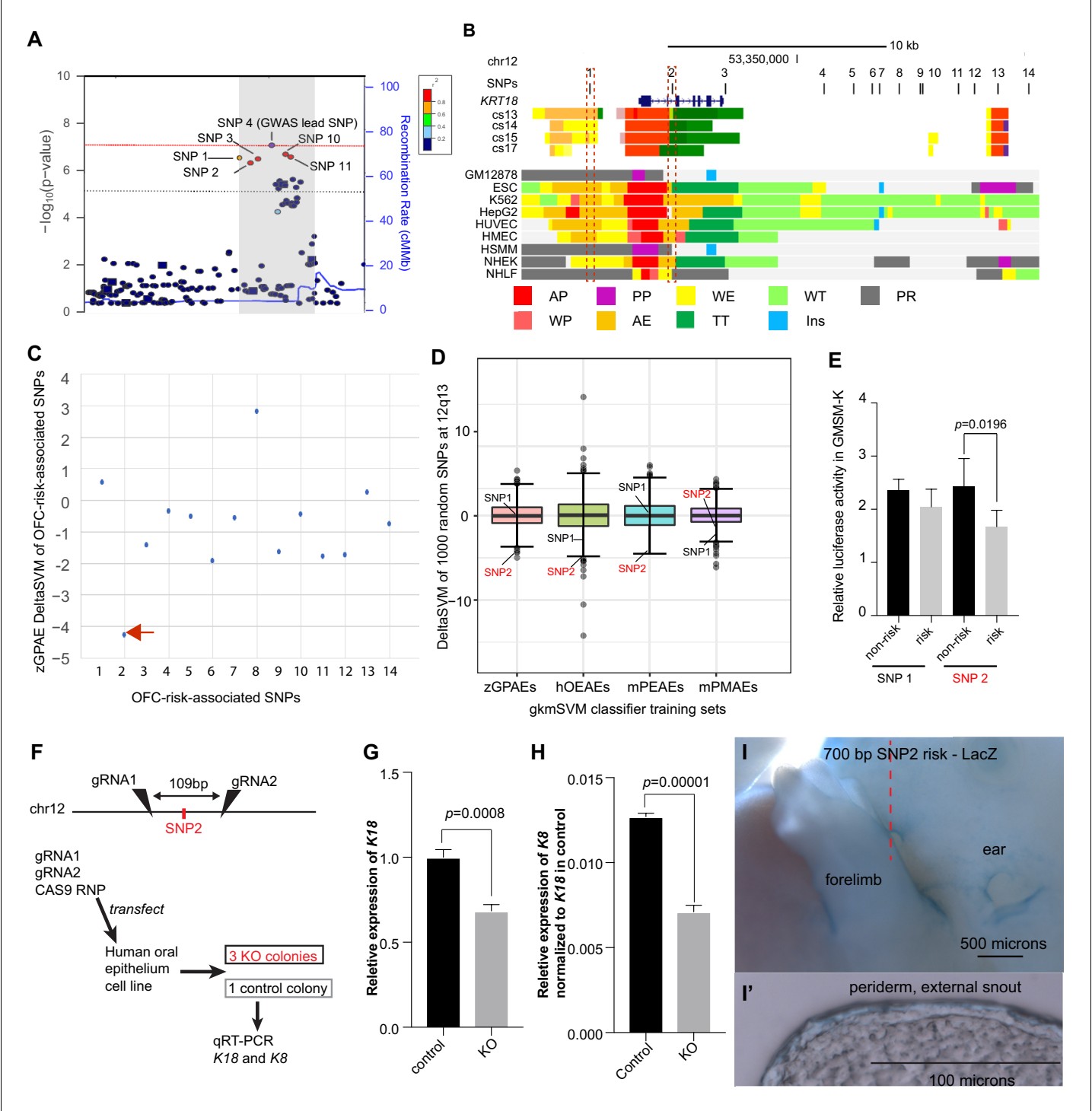

**Figure 6.** Use of a classifier trained on zGPAEs to prioritize orofacial clefting (OFC)-associated SNPs near *KRT18* for functional tests . (**A**) Regional plot showing OFC-risk-associated single nucleotide polymorphism (SNPs) near *KRT18* from this study. SNP4 is the lead SNP from our meta-analysis of OFC GWAS (*Leslie et al., 2017*). (**B**) Browser view of the human genome, hg19, focused on this locus. Tracks: SNPs: OFC-risk-associated SNPs. SNP1: rs11170342, SNP2: rs2070875, SNP3: rs3741442, SNP4: rs11170344, SNP5: rs7299694, SNP6: rs6580920, SNP7: rs4503623, SNP8: rs2363635, SNP9: rs2682339, SNP10: rs111680692, SNP11: rs2363632, SNP12: rs4919749, SNP13: rs2638522, SNP14: rs9634243. Color coded bars: Chromatin status (color code explained in key), revealed by ChIP-seq to various chromatin marks. Cs13-cs17, facial explants from human embryos at Carnegie stage (cs) 13–17, encompassing the time when palate shelves fuse (*Wilderman et al., 2018*). Roadmap Epigenomics Project cell lines (*Visel et al., 2008*): GM12878, B-cell derived cell line; ESC, Embryonic stem cells; K562, myelogenous leukemia; HepG2, liver cancer; HUVEC, Human *umbilical vein endothelial cells*; HMEC, human mammary epithelial cells; HSMM, human skeletal muscle myoblasts; NHEK, normal human epidermal keratinocytes; NHLF, normal

*Figure 6 continued on next page*

*Figure 6 continued*

human lung fibroblasts. AP, active promoter; WP, weak promoter; PP, poised promoter; AE, active enhancer; WE, weak enhancer; TT, transcriptional transition; WT, weakly transcribed; Ins, insulator; PR, polycomb-repressed. (**C**) deltaSVM scores predicted by zGPAEs-derived classifier for the 14 OFC associated SNPs near *KRT1*. (**D**) Box and whisker plots of deltaSVM scores of 1000 randomly-selected SNPs near KRT18, scored by classifiers trained by zGPAEs (zebrafish periderm active enhancers), hOEAEs (human oral epithelium active enhancers), mPEAEs (mouse palatal epithelium active enhancers) and mPMAEs (mouse palatal mesenchyme active enhancers). The line is the median scoring SNP, the box contains the middle-scoring two quartiles, and the whisker represent the top and lower quartiles. Dots are outliers. deltaSVM scores for SNP1 and SNP2 are indicated. Number out of 1000 randomly selected SNPs with a lower deltaSVM than SNP2 with classifier trained on zGPAEs, 2; on mPEAEs, 9; on hOEAEs, 17; on mPMAEs, 186. (**E**) Dual luciferase assay for non-risk and risk alleles of rs11170342 (SNP1) and rs2070875 (SNP2) in GMSM-K cells. (**F**) Schematic diagram showing the workflow of generating GMSM-K cell colonies with 109 bp flanking SNP2 deleted by CRISPR-Cas9. (**G,H**) qRT-PCR showing relative RNA expression of *KRT18* (**G**) and *KRT8* (**H**) in three homozygous knockout colonies (KO) and one isolated wild-type colony (Control) of GMSM-K cell lines. (**I**) Lateral view of transgenic mice LacZ reporter assay for the 700 bp DNA fragment overlapping SNP2. (**I'**) Section of the facial prominence from I (red circled region).

The online version of this article includes the following source data and figure supplement(s) for figure 6:

**Source data 1.** Barchart for relative dual luciferase activity in GMSM-K cells, as plotted in *Figure 6E*.

**Source data 2.** Barchart for relative gene expression of *K18* and *K8* in GMSM-K cells, as plotted in *Figure 6G and H*.

**Figure supplement 1.** Dot plot of deltaSVM scores for each SNP calculated with classifiers trained on the indicated set of enhancer candidates.

**Figure supplement 2.** Bargraphs showing relative RNA expression of K18 (**A**) and K8 (**B**) in GMSM-K cells.

**Figure supplement 3.** Lateral views of all wild-type mouse embryos for *LacZ* reporter assay.

experimental cells, with two gRNAs targeting sites separated by 109 bp and on either side of SNP2, or, in control cells, with a non-targeting gRNA. Two days post-transfection, quantitative RT-PCR on RNA harvested from the pools of cells revealed that *KRT18* mRNA, and at lower levels, *KRT8*, could be detected in control cells, and that levels of both transcripts trended lower in experimental cells (*Figure 6—figure supplement 2*). We isolated single-cells and expanded clones from both the control and experimental cells. Among the latter, we used PCR and sequencing to identify three independent clones that were homozygous for a 109 bp deletion between the two gRNA target sites. The average expression level of *KRT18* and *KRT8* in these three colonies was lower than in a single colony we isolated from the control cells (*Figure 6G,H*). These results support the notion that the region containing SNP2 is an enhancer driving expression of *KRT8* and *KRT18* in human oral epithelial cells.

*KRT18* is expressed in many epithelia other than periderm, including trophectoderm (*Foshay and Gallicano, 2009*), embryonic surface ectoderm (*McGowan and Coulombe, 1998*; *Tadeu and Horsley, 2013*), oral epithelia (whether basal or superficial layer not reported) (*Gong et al., 2005*), embryonic cornea (*Gong et al., 2005*), gonad (*Appert et al., 1998*), bladder (*Erman et al., 2006*), choroid plexus (*Diez-Roux et al., 2011*) and others. To test the prediction that SNP2 lies in a periderm enhancer, we engineered a 700 bp element centered on SNP2, and harboring either the risk or non-risk allele of it, into the GFP reporter vector. We similarly engineered two GFP reporter constructs from 700 bp elements centered on SNP1 with risk or non-risk alleles of it. Unexpectedly, in zebrafish embryos injected with these constructs and monitored up until 4 days post fertilization, we detected very little GFP expression and no consistent pattern of it (N > 100 embryos each construct). This implies either these elements do not function as enhancers in zebrafish embryos, or they do not interact efficiently with the basal promoter in the vector. We considered the possibility that, against our prediction, the elements are enhancers in mammals but not in zebrafish. We engineered all four elements (separately) into reporter vectors with a minimal *Shh* promoter and the *LacZ* gene and carried out transgenic reporter assays in F0 mouse embryos using site-directed transgene integration (*Kvon et al., 2020*). Across 18 transgenic embryos injected with SNP1 element, 11 with the non-risk allele, seven with the risk allele, we did not observe reproducible reporter expression, defined as expression in the same anatomical structure in at least two embryos injected with the same construct, and the majority of transgenic embryos did not show any reporter staining (*Figure 6—figure supplement 3A*). Similarly, in 14 transgenic embryos injected with SNP2 element, seven with each allele, we did not observe reproducible reporter expression, and most showed none at all (*Figure 6—figure supplement 3B*). These results indicate that the elements centered on SNP1 and on SNP2, at least when paired with the basal *Shh* promoter and integrated at this locus, do not reproducibly drive high level reporter activity.

Interestingly, however, a single embryo transgenic for the risk allele of the SNP2 element exhibited mild reporter expression in the periderm of the face and limbs (*Figure 6I,I'*, and *Figure 6—figure supplement 3B*). We hypothesized that this embryo had a higher copy number of the reporter vector than the other embryos. PCR analysis confirmed that this embryo carried 8 copies of the reporter constructs, whereas all other embryos transgenic for SNP2 carried only two copies. This result is consistent with the the prediction that SNP2 lies in a periderm enhancer .

## Allele-specific effects of SNP1 and SNP2 transcription factor binding sites

We used JASPAR to assess the transcription factor consensus binding sites in 19 bp windows centered on SNP1 (rs11170342) or SNP2 (rs2070875). The risk allele of SNP1 strongly reduced the score of Plagl1 and Spz1 sites, and created high-scoring sites for SP4/8/9 and KLF14/15. The risk allele of SNP2 strongly reduced the score of SNAI1/2, NFATC1/2/4, and SIX1/2 sites, and created high-scoring sites for HNF4A/G and NR2F1 (*Supplementary file 3f*). Interestingly, Snai2 is expressed in mouse palate epithelium, and Snai1/Snai2 double mutants exhibit abnormal migration of periderm at medial edge palate epithelium (*Murray et al., 2007*).

## Discussion

Here we undertook an analysis of zebrafish periderm enhancers with two objectives relevant to the genetic underpinnings of orofacial clefting. The first was to learn more about the gene regulatory network (GRN) governing periderm differentiation. Already, the human orthologs of four known elements of this GRN, Irf6, Grhl3, Klf17, and simple epithelium keratins, are implicated in risk for non-syndromic orofacial clefting. Therefore, the orthologs of additional members of this GRN, in particular genes at hubs of the network, are candidates to harbor the mutations that constitute the missing heritability for orofacial clefting. The second objective was to use zebrafish periderm enhancers to train a classifier with which to prioritize SNPs in non-coding DNA for their likelihood of disrupting periderm enhancers. Until recently, identifying enhancers of a given specificity has required testing enhancers individually in reporter assays. Chromatin mark ChIP-seq, and ATAC-seq, which requires fewer cells, have permitted the identification of candidate enhancers in a specific tissue or cell type in large numbers (*Quillien et al., 2017*; *Wilkerson et al., 2019*). However, there is currently no cell line model of human palate periderm. Periderm cells could be isolated from murine embryonic palate shelves using fluorescence-activated cell sorting (FACS) and the *Krt17*-GFP transgenic mouse line (*McGowan and Coulombe, 1998*). However, the zebrafish periderm is far more accessible. While tissue-specific enhancers are rarely strongly conserved between fish and mammals, the faithful performance of tissue-specific enhancers near the human *RET* gene in zebrafish transgenic reporter assays implied that, at least in some cases, enhancers specific for a given tissue are composed of the same transcription factor binding sites in the two clades (*Fisher et al., 2006a*). Ultimately we met both of these objectives.

To identify a set of enhancers active in zebrafish periderm we sorted GFP-positive and GFP-negative cells from *Tg(krt4:gfp)* transgenic embryos at 11 hpf and performed ATAC-seq on both populations. About 5% of all ATAC-seq positive elements had at least 2-fold more ATAC-seq reads in GFP-positive cells. For comparison, ATAC-seq on GFP-positive and GFP-negative cells sorted from *Tg (fli1:gfp)* zebrafish embryos at 24 hpf (*Quillien et al., 2017*), or pro-sensory cells sorted from cochlear ducts in *sox2-EGFP* transgenic mice (*Wilkerson et al., 2019*), revealed about 9% and 15%, respectively, of all ATAC-seq elements to have greater read depth in GFP -positive cells. The difference in the fraction of tissue-specific elements may simply reflect differences in the fold-change-in-ATAC-seq-reads filter applied to such elements. Less than half of the NFRs enriched in GFP-positive cells were marked by H3K27Ac in whole embryo lysates from near the same stage (*Bogdanovic et al., 2012*). This observation is consistent with other studies showing the correlation between nucleosome-free status and H3K27Ac signal it is not absolute; for instance, inactive enhancers can be nucleosome free (*Iwafuchi-Doi et al., 2016*). Genes flanking elements meeting both criteria are strongly enriched for those expressed in periderm. Genes flanking elements meeting only the first criterion (more ATAC-seq reads in GFP-positive cells than GFP-negative cells) were similarly enriched, but to a lesser degree. Periderm-specific NFRs lacking H3K27Ac at 8hpf or 24 hpf but associated with genes expressed in periderm may be enhancers that are active at another stage;

consistent with the existence of such elements, a recent study shows the expression profile in zebrafish periderm changes over time (*Cokus et al., 2019*). Further, 10 of 10 tested elements meeting both criteria and proximal to genes known to be highly and specifically expressed in periderm functioned as periderm enhancers in zebrafish reporter assays. Together these findings support our conclusion that elements with more ATAC-seq reads in GFP-positive versus GFP-negative cells sorted from *Tg(krt4:gfp)* embryos at 11 hpf, and marked with H3K27Ac at 8 hpf and or at 24 hpf, have periderm enhancer (or promoter) activity in embryos at these stages.

Assessment of transcription-factor binding sites enriched in ATAC-seq peaks can yield insights into transcriptional regulatory networks (*Lowe et al., 2019*; *Miraldi et al., 2019*), and did so here for the periderm GRN. First, transcription factors known to bind motifs enriched in the zebrafish periderm enhancer candidates include those previously implicated in periderm development, like Irf6, Grhl3 and Klf17, and novel ones, including C/ebp, Fosl, Gata3 and Tead. Interestingly, orthologs (or, in the case of Klf17, the paralog, Klf4) of each these transcription factors are necessary for mammalian skin development, showing the similarity of the relevant GRNs (e.g., GRHL [*Gordon et al., 2014*; *Nishino et al., 2017*],TEAD-YAP (*Elbediwy et al., 2016*), FOS-JUN/AP-1 (*Uluçkan et al., 2015*), KLF4 (*Segre et al., 1999*), TFAP2 (*Leask et al., 1991*), GATA6 (*Yang et al., 2002*), C/EBP (*Sato et al., 2012*), ETS1 (*Chin et al., 2013*; *Nagarajan et al., 2010*), IRF6 [*Ingraham et al., 2006*]). A role for Tead family members in periderm development is supported by the observation that a loss of function mutation in *yap*, encoding a cofactor of Tead, disrupts periderm development in medaka (*Porazinski et al., 2015*). Knowledge of the key elements of the periderm GRN may help in prioritizing variants that are discovered in patients with orofacial clefting through whole exome or whole genome analyses. Second, whereas Irf6 sites are found in just 5% of enhancer candidates, Grhl sites are present in most. This implies that although both of these transcription factor families are essential for periderm development, Grhl proteins function much more broadly than their Irf6 counterparts. Third, the Irf6 binding site was enriched to a greater extent in the enhancer candidates that were associated with more anti-H3K27Ac ChIP-seq reads at 8 hour post fertilization (hpf) than at 11 hpf. This implies that Irf6 acts early in the GRN; the notion that Irf6 activates a periderm program and is then no longer necessary is consistent with the fact that zygotic *irf6* mutants are viable (*Li et al., 2017*). Finally, the zGPAEs associated with genes encoding candidate members of the GRN contain the binding sites for other such members, implying that mutual cross-regulation of these transcription factor is extensive. While direct connections inferred in this way await confirmation by ChIP-seq or CUT&RUN, cross-regulation among members of a given GRN layer is a common feature in development (*Davidson, 2009*). Of note, ATAC-seq data can be combined with RNA-seq data to yield models of transcriptional regulatory networks that approach the more difficult-to-achieve network models achieved with ChIP-seq and transcription-factor knockout studies (*Miraldi et al., 2019*).

Next, we applied machine-learning classifier to zebrafish periderm enhancer candidates and used it to test the prediction that zebrafish periderm enhancers are enriched for the same sequence motifs as their mammalian counterparts. While gapped kmer support vector machine classifier was the top-performing algorithm at the time we began the study, others are available now that may function better in some circumstances (*Liu et al., 2017b*). The trained classifier had a low false positive rate and auROC and auPR curves comparable to those in similar published studies (*Gorkin et al., 2012*; *Chen et al., 2018*). We applied gkmSVM to tiles of the zebrafish genome. Plotting the average score of tiles overlapping the training set and that of those that do not was useful in conveying the potential of this tool to distinguish periderm enhancers based on sequence alone. On average tiles overlapping biochemically-predicted periderm enhancers (i.e., zGPAEs) have higher scores than those that do not, but plots of these two distributions overlap, indicating that a high score is not a guarantee that an element is a periderm enhancer. We also applied the classifier to the human genome and discovered that enhancers in various epithelial cell types are enriched for high-scoring tilesmore so than other classes of enhancers. This suggests that epithelial enhancers are similarly constructed in these vertebrate organisms. In some cases, high-scoring tiles in the human genome may be orthologs of zebrafish periderm enhancers. An example of such an instance may be a high-scoring tile 8.3 kb upstream of the human *PPL* gene which we found had detectable sequence conservation to an GFP-positive NFR 10 kb upstream of the zebrafish *ppl* gene. Applying the classifier to the murine genome revealed that high-scoring tiles were enriched near genes expressed in periderm, revealing that periderm enhancers in zebrafish and mouse are enriched for the same

sequence motifs. This implies human periderm enhancers share this feature and supports the use of the classifier in deltaSVM analysis of disease-associated SNPs near genes expressed in periderm.

Finally, we used the classifier trained on zGPAEs to prioritize orofacial-cleft associated SNPs near *KRT18,* a gene expressed in periderm, for those that are likely to affect a periderm enhancer. Interestingly, classifiers trained on enhancers apparently selective for a) zebrafish periderm versus non-periderm, b) mouse palate epithelium versus palate mesenchyme, or c) a human oral epithelium cell line versus a human palate mesenchyme cell line all picked SNP2 (i.e., rs2070875) as having the strongest Delta SVM among the 14 OFC-associated SNPs at this locus. This again supports the notion that epithelial enhancers are similarly constructed in all vertebrates. Although the sets of enriched transcription factor binding sites enriched in the three sets of enhancer were similar, only the classifier trained on zebrafish periderm enhancers yielded a deltaSVM for SNP2 that met formal significance. Assuming that SNP2 is indeed functional, these results indicate that the success of deltaSVM in identifying functional SNPs depends more on the enhancers being derived from the correct tissue (i.e., periderm) than the correct species (i.e., human).

Is SNP2 the orofacial-cleft (OFC)-associated SNP at this locus that directly affects risk for the disorder? Luciferase assays in human oral epithelium cells support the notion that SNP1 and SNP2 lie in enhancers active in oral epithelium, and support the predictions of the deltaSVM analysis that SNP2 but not SNP1 affects the activity of the enhancer encompassing them. The fact that homozygous deletion of the 109 bp region flanking SNP2 reduced expression of *KRT18* and *KRT8,* also supports SNP2 affecting OFC risk as periderm integrity is compromised in zebrafish embryos depleted of several keratin genes (*Pei et al., 2007*). Unexpectedly these elements were not consistently active in reporter assays carried out in zebrafish and mouse embryos. However, robust chromatin mark evidence from human embryonic faces and human epidermal keratinocytes suggests the lack ennhancer activity of these elements in the zebrafish and mouse embryo assays reflects a technical artifact; for instance it is possible these enhancers are not compatible with the basal promoter in the *GFP* vector (from the FOS gene *Fisher et al., 2006b*) or in the *LacZ* vector (from the *Shh* gene). In zebrafish embryo reporter assays, enhancers perform more robustly when paired with their cognate promoters than with an exogenous one (*Quillien et al., 2017*). Interestingly, in a single mouse embryo with 8 copies of the SNP2 transgene construct, reporter expression was clearly detected in external periderm. The construct had the risk allele of SNP2, but as reporter copy number varied among embryos, we could not assess whether the allele affected reporter level in this assay. Generating embryos with multiple integrated copies of the constructs, or perhaps targeting the constructs to a different locus, will be necessary to determine if the enhancer harboring SNP2 is active in oral periderm as would be predicted if SNP2 is relevant to risk for orofacial clefting. In summary, the data gathered support SNP2 as affecting expression of an enhancer active in periderm and regulating *KRT18* and *KRT8* expression.

We conclude by suggesting the classifiers presented may be useful in nominating functional SNPs at the additional loci identified in orofacial clefting genome wide association studies. At loci where the candidate risk gene is expressed in oral epithelium, for example *IRF6, MAFB, FOXE1, TP63,* the classifiers trained on zGPAEs, mPEAEs, and hOEAEs should all be applied; if the risk gene is expressed in basal epithelium, like *TP63,* the classifier trained on mPEAEs may work better than the one trained on zGPAEs. Where the candidate risk gene is expressed in mesenchyme (e.g., *PAX7*), the classifier trained on mPMAEs is expected to be the most accurate. It is important to note that in a study of over 100 SNPs, the correlation between deltaSVM scores and the effects of SNPs on reporter level (in an appropriate cell type) was significant but modest (*Lee et al., 2015*). Therefore, machine learning analyses can prioritize SNPs but functional tests remain essential. Such tests include quantification of a SNP's effects on enhancer activity, either by reporter assays in vitro, or more powerfully, through genome engineering of an appropriate cell line to render the SNP homozygous for then risk or non-risk allele and then quantification of RNA levels for the proximal risk-relevant gene. The efficiency of homology directed repair for this purpose remains highly locus-dependent, and we did not succeed in applying it here, although we did so at another locus (*Liu et al., 2017a*). Fortunately, single-nucleotide editing tools are improving rapidly (*Zafra et al., 2018*; *Anzalone et al., 2019*). Finally, in vivo reporter assays will remain essential for testing the tissue specificity of enhancers harboring the candidate functional SNPs. In such assays, safe harbor chromatin integration is clearly desired, although such safe harbors may be more permissive for

expression in some tissues than others, and enhancer-promoter compatibility may additionally affect efficiency of in vivo reporter assays (*Quillien et al., 2017*).

# Materials and methods

## Key resources table

| Reagent type (species) or resource | Designation | Source or reference | Identifiers | Additional information |
|---|---|---|---|---|
| Strain, strain background (*Escherichia coli*) | One Shot TOP10 | Life technologies | Cat# C4040-10 | Chemically competent cells |
| Cell line (*Homo-sapiens*) | GMSM-K (human embryonic oral epithelial cell line) | (*Gilchrist et al., 2000*) | RRID:CVCL_6A82 | a kind gift from Dr. Daniel Grenier |
| Cell line (*Homo-sapiens*) | HIOEC (human immortalized oral epithelial cells) | (*Sdek et al., 2006*) | RRID:CVCL_6E43 | |
| Cell line (*Homo-sapiens*) | HEPM (human embryonic palatal mesenchyme cells) | ATCC | ATCC Cat# CRL-1486, RRID:CVCL_2486 | |
| Antibody | anti-Histone H3, Acetylated Lysine 27 (Rabbit polyclonal) | Abcam | Abcam Cat# ab4729, RRID:AB_2118291; lot NO. GR3211959-1; | ChIP (4 ug per 500,000 HIOEC cells) |
| Recombinant DNA reagent | pXX330 (plasmid) | Addgene; {Cong, 2013 #2; Ran, 2013 #1} | RRID :Addgene_ 42230 | |
| Recombinant DNA reagent | cFos-GFP | (*Fisher et al., 2006b*) | | a gift from Shannon Fisher |
| Recombinant DNA reagent | cFos-tdTomato | This paper | | Modified from cFos-GFP |
| Recombinant DNA reagent | pENTR/D-TOPO | Life technologies | Invitrogen Cat# K240020 | |
| Recombinant DNA reagent | cFos-FFLuc | (*Liu et al., 2017a*) | | |
| Sequence-based reagent | cFos-RLuc | (*Liu et al., 2017a*) | | |
| Sequence-based reagent | Klf17_+1.8_F | This paper | PCR primers | ATGCTGACTCCA CCATCCTC |
| Sequence-based reagent | Klf17_+1.8_R | This paper | PCR primers | CACCTACCCCTTGGC TAATCGTTG |
| Sequence-based reagent | Cavin2b_+18_F | This paper | PCR primers | TTCTGTTTTTGC CATCAGCA |
| Sequence-based reagent | Cavin2b_+18_R | This paper | PCR primers | CACCTTTTAATCAC CGCCTTTCCA |
| Sequence-based reagent | Gadd45ba_−0.7_F | This paper | PCR primers | TGGTTGGGTTC AGAGGTAGG |
| Sequence-based reagent | Gadd45ba_−0.7_R | This paper | PCR primers | CACCATGACTCGAC GAAAGCAAA |
| Sequence-based reagent | SNP2_gRNA_left | This paper | gRNA target | CTAAGAAGGATC TGCTCCCC |
| Commercial assay or kit | SNP2_gRNA_right | This paper | gRNA target | GAGGACAGTATTC TTAAACG |
| Commercial assay or kit | RNAqueous Total RNA Isolation Kit | Ambion | Cat# AM1912 | |
| Commercial assay or kit | RNA Clean and Concentrator-5 Kit | Zymo Research | Cat# R1013 | |

*Continued on next page*

*Continued*

| Reagent type (species) or resource | Designation | Source or reference | Identifiers | Additional information |
|---|---|---|---|---|
| Commercial assay or kit | SMART-Seq v4 Ultra Low Input RNA Kit | TAKARA | Cat# 634888 | |
| Commercial assay or kit | Agilent RNA 6000 Pico | Agilent Technologies | Cat# 5067–1513 | |
| Commercial assay or kit | Nextera XT DNA Sample Preparation Kit | Illumina | Cat# FC-131–1002 | |
| Commercial assay or kit | Nextera DNA Sample Preparation Kit | Illumina | Cat# FC-121–1030 | |
| Commercial assay or kit | VAHTS Universal DNA Library Prep Kit for Illumina | Vanzyme | Cat# ND606-01 | |
| Commercial assay or kit | KAPA Library Quantification Kit | Roche | Cat# KK4824 | |
| Commercial assay or kit | NEBNext High-Fidelity 2x PCR Master Mix | New England Biolabs | Cat# M0541S | |
| Chemical compound, drug | Ampure XP beads | Beckman Coutler | Cat# A63881 | |
| Chemical compound, drug | 0.25% trypsin-EDTA | Life Technologies | Cat# 25200056 | |
| Chemical compound, drug | Defined trypsin inhibitor | Life Technologies | Cat# R007100 | |
| Software, algorithm | Turbo DNase I | Ambion | Cat# AM2238 | |
| Software, algorithm | R | R | RRID:SCR_001905 | v 3.5.1 v 3.3.2 |
| Software, algorithm | Bowtie2 | (*Langmead and Salzberg, 2012*) | RRID:SCR_005476 | v 2.3.4.1 |
| Software, algorithm | Trimmomatic | (*Bolger et al., 2014*) | RRID:SCR_011848 | v.0.38 |
| Software, algorithm | DiffBind | (*Ross-Innes et al., 2012*) | RRID:SCR_012918 | |
| Software, algorithm | seqMINER | (*Ye et al., 2011*) | RRID:SCR_013020 | v 1.2.1 |
| Software, algorithm | HOMER | (*Heinz et al., 2010*) | RRID:SCR_010881 | v 3.0 |
| Software, algorithm | Gapped k-mer support vector machine | (*Ghandi et al., 2016*) | https://rdrr.io/cran/gkmSVM/ | v 0.79.0 |
| Software, algorithm | BEDTools | (*Quinlan and Hall, 2010*) | RRID:SCR_006646 | v 2.24.0 |
| Software, algorithm | Picard Tools | http://broadinstitute.github.io/picard/ | RRID:SCR_006525 | v 0.35 |
| Software, algorithm | SAMtools | (*Li et al., 2009*) | RRID:SCR_002105 | v 1.7 |
| Software, algorithm | MACS2 | (*Zhang et al., 2008*) | RRID:SCR_013291 | v 2.1.1 |
| Software, algorithm | DeepTools | (*Ramírez et al., 2016*) | RRID:SCR_016366 | v 2.0 |

## Identification of SNPs associated with human orofacial clefting

We re-analyzed our published meta-analysis of two GWASs for orofacial clefting (*Leslie et al., 2017*).The details of each contributing GWAS have been extensively described. Data are from a total of 823 cases, 1700 controls, and 2811 case-parent trios and were obtained by genotyping using the Illumina HumanCore+Exome array or the Illumina Human610-Quad array. In our re-analysis, genotype probabilities for imputed SNPs were converted to most-likely genotype calls using GTOOL (http://www.well.ox.ac.uk/~cfreeman/software/gwas/gtool.html). Genotype calls were retained for analysis only if the genotype with the highest probability was greater than 0.9. SNPs were excluded if the minor allele frequency was low (<1%), the imputation quality scores was low (INFO <0.5) or deviating from Hardy-Weinberg equilibrium in genetically defined Europeans (p<0.0001). Statistical analyses were performed as previously described (*Leslie et al., 2017*), using an inverse variance-

weighted fixed-effects meta-analysis. At the 12q13.13 locus, we prioritized 14 SNPs whose p-values indicated strong linkage disequilibrium (<1E-5) (listed in *Supplementary file 1j*). These included two SNPs with p-values reaching formal genome-wide significance (5E-8): rs11170344, the lead SNP in our re-analysis; and rs3741442, which was identified in a recent GWAS in a Chinese orofacial cleft population. (*Yu et al., 2017*).

## Zebrafish lines and maintenance

*D. rerio* were maintained in the University of Iowa Animal Care Facility according to a standard protocol (protocol no. 6011616). (*Westerfield, 1993*) All zebrafish experiments were performed in compliance with the ethical regulations of the Institutional Animal Care and Use Committee at the University of Iowa and in compliance with NIH guidelines. Zebrafish embryos were maintained at 28.5℃, and staged by hours or days post-fertilization (hpf or dpf).

## Mouse maintenance

All C57BL/6 mouse experiments used for ATAC-seq library preparation were performed in accordance with approval of the Institutional Animal Care and Use Committees at the School and Hospital of Stomatology of Wuhan University (protocol no. 00271454). Mouse experiments for *LacZ* reporter transgenic animal work performed at the Lawrence Berkeley National Laboratory (LBNL) were reviewed and approved by the LBNL Animal Welfare and Research Committee. Transgenic mouse assays were performed in *Mus musculus* FVB strain mice.

## Cell culture

GMSM-K human embryonic oral epithelial cell line (a kind gift from Dr. Daniel Grenier; *Gilchrist et al., 2000*) and the human immortalized oral epithelial cell (HIOEC) line (*Sdek et al., 2006*) were maintained in keratinocyte serum-free medium (Life Technologies, Carlsbad, CA) supplemented with EGF and bovine pituitary extract (Life Technologies). HEPM human embryonic palatal mesenchyme cells (ATCC CRL-1486) were maintained in DMEM (Hyclone, Pittsburgh, PA) supplemented with 10% fatal bovine serum (Hyclone). All the cell lines used in this study were tested for mycoplasma contamination and authenticated by genetic profiling using polymorphic short tandem repeats.

## Electroporation and dual luciferase assay

For dual luciferase assays, each reporter construct was co-transfected with Renilla luciferase plasmid and three biological replicates were used. Briefly, GMSM-K cells were electroporated with plasmid using the Amaxa Cell Line Nucleofector Kit (Lonza, Cologne, Germany) and the Nucleofector II instrument (Lonza). We used a dual-luciferase reporter assay system (Promega, Madison, WI) and 20/20 n Luminometer (Turner Biosystems, Sunnyvale, CA) to evaluate the luciferase activity following manufacturer's instructions. Relative luciferase activity was calculated as the ratio between the value for the Firefly and Renilla enzymes. Three independent measurements were performed for each transfection group. All results are presented as mean ±s.d. Statistical significance was determined using the Student's *t*-test.

## Plasmid constructs and transient reporter analysis of potential periderm enhancer in zebrafish and mouse

All candidate enhancer elements described were cloned using zebrafish or human genomic DNA, and were harvested from either zebrafish embryos or a human immortalized oral keratinocyte cell line (GMSM-K; *Gilchrist et al., 2000*). Products were cloned into the pENTR/D-TOPO plasmid (Life Technologies, Carlsbad, CA) and validated by Sanger sequencing. Site-directed mutagenesis was used to generate elements lacking the corresponding motifs or containing a risk variant. All elements were subcloned into the *cFos-GFP* plasmid (a gift from Shannon Fisher; *Fisher et al., 2006b*) or *cFos-tdTomato*, a derivative of *cFos-GFP*, or *cFos-GFP; Cry-GFP*, a derivative that includes a lens-specific promoter (cloning details available upon request). For each reporter construct, at least 100 embryos at the 1 cell to 2 cell stage were injected (20 pg reporter construct plus 20 pg tol2 mRNA); three replicates were performed, each on a different day (*Fisher et al., 2006b*). Embryos injected were examined by epifluorescence microscopy first at approximately 11 hpf, then each day

subsequently until approximately four dpf. SNP1 and SNP2 700 bp elements, used in GFP and *LacZ* reporter constructs were (SNP1) chr12:53,340,250–53,340,950 and (SNP2) chr12:53,343,968–53,344,668 (hg19). Specifically, for mouse reporter assay, candidate enhancers were PCR-amplified and cloned upstream of a *Shh*-promoter-*LacZ*-reporter cassette. We used a mouse enhancer-reporter assay that relies on site-specific integration of a transgene into the mouse genome (*Kvon et al., 2020*). In this assay, the reporter cassette is flanked by homology arms targeting the H11 safe harbor locus (*Tasic et al., 2011*). Cas9 protein and a sgRNA targeting H11 were co-injected into the pronucleus of FVB single cell stage mouse embryos (E0.5) together with the reporter vector (*Kvon et al., 2020*). Embryos were sampled and stained at E13.5. Embryos were only excluded from further analysis if they did not carry the reporter transgene. Transgene copy number was estimated by qPCR using a TaqMan probe targeting *Shh* promoter.

## Dissociation of zebrafish embryos and FACS

About 500 *Tg(krt4:GFP)* (*Gong et al., 2002*) embryos were collected at the 4-somite stage and rinsed with PBS without $Ca^{2+}$ or $Mg^{2+}$ (Life Technologies). Embryos were dechorionated using pronase (Sigma, St. Louis, MO, 1 mg/mL in fish water 1 mg/mL in fish water) at room temperature for 10 min, rinsed in PBS, and then dissociated cells using a pestle and incubated in trypsin (0.25%)-EDTA (Life technology) at 33°C for 30 min. Reactions were stopped by adding PBS supplemented with 5% fetal bovine serum (Life technologies). Dissociated cells were re-suspended into single-cell solution and analyzed at the University of Iowa Flow Cytometry Facility, using an Aria Fusion instrument (Becton Dickinson, Franklin Lakes, NJ).

## Dissociation of mouse palatal epithelium

Mouse embryos at were collected at E14.5 and palate shelves were dissected. The multi-layered palate shelf epithelium was manually isolated as described previously (*Zhang et al., 2017*). Briefly, palatal shelves isolated from the frontal facial prominence were incubated in 0.25% trypsin-EDTA (Life Technologies) at 4°C for 10 min, after which the reaction was stopped using Trypsin Inhibitor (Life Technologies). Under a dissecting microscope, the epithelium was isolated for ATAC-seq by gently peeling using microforceps. The remaining tissue, comprised largely of mesenchymal cells, was also collected for control samples. This isolation protocol was previously described as a method for harvesting oral periderm (*Zhang et al., 2017*), but in our experiment it was clear that the epithelial layers were harvested together. Supporting the basal epithelium being harvested with the epithelial and not the mesenchymal layer, the ATAC-seq results from the former and not the latter show open chromatin at the *Tp63* gene. Approximately 20,000 epithelial cells and the same number of mesenchymal cells were harvested from seven embryos at E14.5. About 20,000 cells were used to prepare one ATAC-seq library.

## Preparation of RNA-seq libraries and high-throughput sequencing

Three independent biological replicates were subjected to RNA-seq profiling. In each replicate, we isolated 20,000 peridermal and non-peridermal cells from *Tg(krt4:GFP)* embryos at the 4-somite stage. Total RNA was extracted from the sorted cells using the RNAqueous Total RNA Isolation Kit (Ambion, Foster City, CA) and treated with Turbo DNase I (Ambion, Austin, TX) to remove residual genomic DNA. The treated RNA was then purified and concentrated using the RNA Clean and Concentrator-5 Kit (Zymo Research, Irvine, CA). After quantification with Qubit 3.0 (Life Technologies) and quality control with the Agilent RNA 6000 Pico Kit on Agilent 2100 (Agilent Technologies, Santa Clara, CA), 1 ug of RNA was subjected to first-strand cDNA synthesis and cDNA amplification using the SMART-Seq v4 Ultra Low Input RNA Kit (Takara Bio, Kusatsu, Shiga, Japan). Purified cDNA was quantified using Qubit (Life Technologies), and for each library 150 pg cDNA was used as input with the Nextera XT DNA Sample Preparation Kit (Illumina, San Diego, CA), following the manufacturer's instructions. Each DNA library was quantified using a KAPA Library Quantification Kit (Roche, Mannheim, Germany) and pooled for HiSeq4000 (Illumina) high-throughput sequencing at the same molarity.

## RNA-seq data analysis

RNA-seq raw reads data was trimmed using the Trimmomatic (Usadel Lab, Aachen, Germany. v0.36; *Bolger et al., 2014*) (parameter: ILLUMINACLIP:TrueSeq3PE-PE.fa:2:30:10:8:TRUE LEADING:3 TRAILING:3 SLIDINGWINDOW:4:15 MINLEN:30), and aligned to zv10 cDNA reference sequence using the kallisto (Pachter Lab, California Institute of Technology, Pasadena, CA) aligner (*Bray et al., 2016*) with the default parameters. The output from kallisto was quantified using the sleuth R package (Pachter Lab; *Pimentel et al., 2017*) according to a standard protocol. We used q-value <0.01 and p-value<0.01 as the threshold for significant difference. Gene set enrichment analysis was performed using GSEA (v 3.0) (*Subramanian et al., 2005*). Gene ontology (GO) enrichment analysis was performed using the Metascape online tool (http://metascape.org/gp/index.html) (*Tripathi et al., 2015*), and the top GO categories were selected according to the binomial *P* values. Raw and processed sequencing data for RNA-seq were deposited in GEO repository (GSE140241).

## Chromatin immunoprecipitation of H3K27 acetylation (H3K27Ac) combined with high throughput sequencing (ChIP-seq)

HIOEC cells were seeded at $1 \times 10^5$ cells per 100 mm plate (one plate per biological replicate), grown to 90–100% confluency (refreshed medium every other day) and subjected to 1.2 mM $Ca^{2+}$ in culture medium for 3 days. Cell were washed with ice-cold PBS (Hyclone) and fixed with 1% paraformaldehyde (PFA) for 10 min at room temperature. PFA was then quenched in 134M Glycine (Sigma) for 5 min at room temperature, and cells were collected with a cell scraper in ice-cold PBS. After centrifuge for 10 mins at 500 g, the cell pellets were resuspended with 5 mM PIPES pH8.5, 85 mM KCl, 1% (v/v) IGEPAL CA-630, 50 mM NaF, 1 mM PMSF, 1 mM phenylarsine oxide, 5 mM Sodium Orthovanadate and protease inhibitor cocktail (Roche, Germany). After sonication, chromatin immunoprecipitation was performed using 4 ug of anti-Histone H3, Acetylated Lysine 27 (H3K27Ac) (Abcam, Cambridge, UK, ab4729, lot NO. GR3211959-1) per 500,000 cells. ChIP-seq library were indexed with kit (ND606-01, Vanzyme, China). 150-bp-paired-end sequencing was performed using the HiSeq X Ten sequencer (Illumina, provided by Annoroad Genomics, China). Output sequences were trimmed using Trimmomatic (v0.38) (*Bolger et al., 2014*) (parameter: ILLUMINACLIP:TrueSeq3PE-PE.fa:2:30:10:8:TRUE LEADING:3 TRAILING:3 SLIDINGWINDOW:4:15 MINLEN:30). All trimmed, paired reads were aligned to human genome assembly 19 (hg19) using Bowtie 2 (Johns Hopkins University, Baltimore, MD, default parameters; *Langmead and Salzberg, 2012*). Peaks were called using MACS2 (v2.1.1) (*Zhang et al., 2008*). DeepTools (Max Planck Institute for Immunobiology and Epigenetics, Freiburg, Germany, v 2.0) was used to confirm reproducibility of the biological replicates and generate bigWig coverage files for visualization (*Ramírez et al., 2016*). Raw and processed sequencing data for this H3K27Ac ChIP-seq were deposited in GEO repository (GSE139809).

## Preparation of ATAC-seq library and high-throughput sequencing

We prepared the ATAC-seq library according to a previously published protocol (*Buenrostro et al., 2013*). Briefly, sorted cells were lysed with 50 µL cold lysis buffer (10 mM Tris--HCl, pH 7.4, 10 mM NaCl, 3 mM MgCl2, 0.1% NP-40; all components purchased from Sigma) in a centrifuge at 500 x g for 15 min at 4˚C. Pelleted nuclei were resuspended in 50 µL tagmentation reaction mix (25 µL Nextera TD Buffer, 2.5 µL Nextera TD Enzyme, and 22.5 µL $H_2O$, all from the Nextera DNA Sample Preparation Kit [Illumina]). Tagmentation was performed at 37˚C for 30 min in a thermocycler and, immediately after the reaction was completed, the DNA was purified using a Qiagen PCR Purification MinElute Kit (QIAGEN, Germantown, MD) and eluted with 10 µL elution buffer. Eluted DNA was subjected to PCR amplification and library indexing, using the NEBNext High-Fidelity 2x PCR Master Mix (New England Biolabs, Ipswich, MA) with a customized Nextera PCR primer pair, according to the following program: 72˚C for 5 min; 98˚C for 30 s; 11 cycles of 98˚C for 10 s, 63˚C for 30 s, and 72˚C for 1 min; and hold at 4˚C. The PCR product was purified with 1.8 x volume (90 µL for each sample) of Ampure XP beads (Beckman Coulter, Brea, CA) to produce 18 µL of final library. Library quality was assessed using 1 µL of the final purified DNA on a BioAnalyzer 2100 High Sensitivity DNA Chip (Agilent Technologies). All DNA libraries that exhibited a nucleosome pattern in the BioAnalyzer 2100 Assay passed the pre-sequencing QC process and were pooled for high-throughput

sequencing in HiSeq 2500, HiSeq4000, or HiSeq X Ten (Illumina, provided by Annoroad Genomics Company (China)).

## Mapping of ATAC-seq reads and calling of peaks and differential peaks

Raw ATAC-seq reads were trimmed using Trimmomatic (v 0.38) (*Bolger et al., 2014*) (parameter: ILLUMINACLIP:NexteraPE-PE.fa:2:30:10:8:TRUE LEADING:3 TRAILING:3 SLIDINGWINDOW:4:15 MINLEN:5) and mapped to either the danRer7, hg19 or mm10 reference genome using Bowtie 2 (*Langmead and Salzberg, 2012*) (default parameters). Sorting, removal of PCR duplicates and conversion from SAM to BAM files were performed using SAMtools (*Li et al., 2009*). A customized Python script was used to identify fragments shorter than 100 bp as the nucleosome-free-regions (NFRs), as previously described (custom scripts and piplines we deployed are available at https://github.com/Badgerliu/periderm_ATACSeq; copy archived at https://github.com/elifesciences-publications/periderm_ATACSeq) (*Buenrostro et al., 2013*). The Picard toolset (http://broadinstitute.github.io/picard/) was used to check fragment size distribution. We employed DeepTools (v 2.0) to check the reproducibility of the biological replicates and generate bigWig coverage files for visualization (*Ramírez et al., 2016*). Peaks were called using MACS2 (v2.1.1) (*Zhang et al., 2008*) (parameter: –nomodel–nolambda –gsize 1.4e9 –keep-dup all –slocal 10000 –extsize 54 for zebrafish periderm ATAC-seq, and –nomodel –nolambda –gsize 2.7e9 –keep-dup all –slocal 10000 for mouse periderm or HOIEC ATAC-seq). Differentially accessible NFRs were identified using an R package DiffBind (*Ross-Innes et al., 2012*) with a fold-change threshold of 0.5, and FDR < 0.01.

Raw and processed sequencing data for zebrafish, mouse and human ATAC-seq were deposited in GEO repository (GSE140241, GSE139945 and GSE139809).

## Integration of ATAC-seq and H3K27Ac ChIP-seq data

To compare our zebrafish periderm ATAC-seq results to those in previously published whole-embryo H3K27Ac ChIP-seq studies (*Bogdanovic et al., 2012*), we retrieved the single-end, raw read data from GEO Series GSE32483 H3K27Ac ChIP-seq from whole zebrafish embryos at several stages. Raw data were aligned to the danRer7 reference genome using Bowtie 2, and peaks were called using MACS2. Following an earlier study (*Gorkin et al., 2012*), we defined 'H3K27Ac-flanked' regions as those between adjacent H3K27Ac peaks separated by up to 1500 bp. NFRs were identified in cells isolated at 11 hpf; GFP-positive active elements (GPAEs) were the GFP-positive NFRs that overlap H3K27Ac peaks, or H3K27Ac flanked regions, at 8.3 hpf (80% epiboly) and/or at 24 hpf embryos; GNAEs were the GFP-negative NFRs that did so.

The approach used to identify enhancers that are active in the mouse palate epithelium was similar. It involved integrating mouse palate-epithelium-specific NFRs found in this study with previously published H3K27Ac ChIP-seq data for E14.5 mouse embryonic facial prominences (GSE82727) (*ENCODE Project Consortium, 2012*). We also identified enhancers that are active in HIOECs by integrating HIOEC-enriched NFRs with the H3K27Ac ChIP-seq results from this study.

We used seqMINER (v 1.2.1) (*Ye et al., 2011*) to calculate the normalized reads matrix for each NFR of interest, generating a matrix file for the downstream heatmap and density plot in R.

## Assignment of ATAC-seq peaks to genes and gene ontology analysis

The Genomic Regions Enrichment of Annotations Tool (GREAT, http://great.stanford.edu/public/html/) (*McLean et al., 2010*) was employed to assign genes proximal to genomic regions of interest (i.e., ATAC-seq and high-scoring elements), using the following locus-to-gene association rule: two nearest genes within 100 kb. We also used GREAT to identify GO terms for which the set of associated genes was enriched.

## Comparisons of peak accessibility and gene expression

To compare ATAC-seq accessibility and relative gene expression between tissues, we first identified genes for which zebrafish peridermal and non-peridermal tissue was enriched using Sleuth as described above. We then identified the normalized ATAC-seq accessibility using the EdgeR package embedded in the DiffBind analysis suite (cutoff: fold change >0.5 or<−0.5, p-value<0.01). We used GREAT to associate the periderm- and non-periderm-enriched genes with their tissue-specific ATAC-seq peaks. To determine whether the accessibility of periderm-specific ATAC-seq correlates

with gene expression in the periderm, we compared the accessibility of tissue-specific ATAC-seq peaks (values normalized) within genes for which either peridermal or non-peridermal tissue is enriched, as well as the levels of expression of genes associated with periderm or non-periderm ATAC-seq peaks.

## Motif enrichment analysis and footprinting for periderm-enriched motifs

We identified the de novo motifs for which the genomic regions of interest are enriched using the findMotifsGenome.pl function of HOMER (*Heinz et al., 2010*) (parameter: -len 8,10,12), and assigned the most enriched motifs to the transcription factors with highest expression in related tissues. For the Tn5 footprint analysis, we shifted all reads aligned to the plus strand by +4 bp, and all reads aligned to the minus strand by −5 bp. To predict the binding of members of the GRHL, KLF, TFAP2, and C/EBP transcription factor families, we downloaded the related motifs in all transcription factors of interest (http://cisbp.ccbr.utoronto.ca/) (*Weirauch et al., 2014*), and calculated the Tn5 cleavage frequency in the +/- 100 bp sequence flanking the motifs of interest, using CENTEPEDE (*Pique-Regi et al., 2011*).

For analysis of the potential clustering pattern of periderm-enriched motif combination within zebrafish GPAEs, we firstly annotated all the GPAEs using HOMER annotatePeaks function with the motif files for GRHL, TEAD, KLF, FOS, TFAP2, GATA and CEBP, and counted the occurrence of each motif in each peak. Hierarchy clustering was then performed on the occurrence of different motif in each peak using the 'hierarchy_cluster_motif_combination_pattern_in_GPAE.R' script deposited in periderm_ATACSeq github repository. We also counted the sum of every two-motif-combination and three-motif-combination in each GPAE using 'motif_combination_count.R' script deposited in github.

## Comparison of TFBS enrichment

To determine the number of binding sites that would have been shared by chance, we generated 10 sets of 4000 randomly-selected 400 bp sequences from two species and assessed the average number of transcription factors predicted to bind sequences enriched in both species. Transcription factors receiving a score of 0.8 in HOMER output (*Heinz et al., 2010*) were considered to match the binding site.

## Construction of network depicting the regulatory relationships among periderm-enriched transcription factors

To assign periderm signature motifs to the periderm-enriched genes, we first annotated all periderm-specific NFRs with signature motifs using HOMER, then calculated the total number of times each motif was present in all of periderm-specific NFRs near the periderm-enriched transcription factor. Expression levels of each transcription factor in the periderm were also taken into account.

## Training of a gapped k-mer support vector machine on zebrafish periderm enhancers

All GFP-positive active enhancers (GPAEs) were resized into 400 bp regions that maximize the overall ATAC-seq signal within each NFR; all GPAEs composed of > 70% repeats were removed. Repeat fractions were calculated using repeat masked sequence data (danRer7) from the UCSC Genome Browser (http://genome.ucsc.edu/). A supervised-machine-learning classifier, gapped k-mer support vector machine (gkmSVM) was used to generate a 10-fold larger set of random genomic 400 bp sequences in the danRer7 reference genome, based on matching of GC and repeat fraction of the positive training set. gkmSVM were performed to generate a scoring vector (parameter: K = 6, L = 10). Related ROC and PRC were generated using gkmSVM output (*Ghandi et al., 2016*). For genome-wide enhancer predictions, mouse (mm10) and human (hg19) genomes were segmented into 400 bp regions with 300 bp overlap, and all regions were scored using a gkmSVM script.

## Tests of enhancer homology

The following DNA fragments were used to test homology of the human and zebrafish enhancers. The 489 bp sequence corresponding to the human *ppl* periderm enhancer (plus-strand) was trimmed

to a conserved 400 bp core block ('Hu_400+'), which lacks an overlapping non-conserved AluJr4 SINE on the *ppl*-proximal side and a non-conserved MIR SINE on the *ppl*-distal side. The homologous mouse sequence to the human 400 bp core enhancer was determined to be a 409 bp fragment ('Mm_409+'). A zebrafish 467 bp core fragment ('Zf_467+') was identified to correspond to be the block most similar to the mammalian *ppl* enhancer core. All three core fragments lie 8.5 kb (human), 4.0 kb (mouse), and 10.4 kb (zebrafish) upstream of *ppl*, which is transcribed to the left in each case. To evaluate homology between Hu_400+ and Zf_467+ enhancer fragments we performed pairwise alignments between various sequences using CLUSTALW and default parameters as available via Clustal Omega (https://www.ebi.ac.uk/Tools/msa/clustalo/). The Hu_400+ | Mm_409+ pairwise alignment constituted the homologous control to compare to the Hu_400+ | Zf_467+ test alignment. For different types of negative controls, we also aligned the Hu_400+ sequence to the mouse and fish sequences corresponding to the minus-strand (i.e., reverse-complements 'Mm_409-' and 'Zf_467-'), the non-biological reverse sequence ('Mm_409R' and 'Zf_467R'), and three different Fisher-Yates shuffled versions of the mouse and zebrafish plus-strand sequences ('Mm_409+S1/S2/3' and 'Zf_467+S1/S2/S3'). Last, we also compared different 3-way alignments involving Hu_400+, Mm_409+, and the various zebrafish test and controls (*Supplementary file 2a* and *Supplementary file 2b*).

## Annotation of potential zebrafish periderm enhancer candidates using ENCODE/Roadmap data

All mapped data from the Roadmap Epigenomics Project (*Kundaje et al., 2015*) (http://www.roadmapepigenomics.org/) were downloaded as BAM files; imputed enhancer regions in each cell/tissue type were also downloaded. Using the BEDTools (v2.24.0) (*Quinlan and Hall, 2010*) intersect function, we evaluated the fraction of high-scored elements that overlapped with enhancer regions in each cell/tissue type.

## Analysis of transcription factor binding sites affected by alleles of SNP1 and SNP2

We used 19 bp sequences centered on the SNP1 and SNP2, with risk or non-risk alleles of these SNPs, as input to JASPAR (http://jaspar.genereg.net) (*Fornes et al., 2019*). We queried all 1011 transcription factor binding site profiles using a relative profile score threshold of 80%. 'Sites lost' were those at a particular start position with a score of 5.0 or higher in the sequence with the non-risk allele and with a score less than 2.0, or not detected, in the sequence with the risk allele. 'Sites gained' had the opposite pattern.

## Acknowledgements

This work supported by grants from the NIH including DE023575 (RAC), DE027362 (RAC), DE025060 (EJL), DE024427 (AV), DE028599 (AV), and NCI P30CA086862 (University of Iowa Flow Cytometry Facility), from the National Natural Science Foundation of China (NO. 81771057; 81400477) (HL), the Natural Science Foundation of Hubei Province (NO. 2017CFB515) (HL) and the Young Elite Scientist Sponsorship Program by CAST (NO. 2017QNRC001) (HL). Research at Lawrence Berkeley National Laboratory was performed under Department of Energy Contract DE-AC02-05CH11231, University of California. The numerical calculations in this paper have been done on the supercomputing system in the Supercomputing Center of Wuhan University and at the High Performance Computing Cluster at the University of Iowa.

## Additional information

### Funding

| Funder | Grant reference number | Author |
| --- | --- | --- |
| National Institutes of Health | DE023575 | Robert A Cornell |
| National Institute for Health Research | DE027362 | Robert A Cornell |

| National Institute of Dental and Craniofacial Research | DE025060 | Elizabeth Leslie |
|---|---|---|
| National Institute of Dental and Craniofacial Research | DE024427 | Axel Visel |
| National Institute of Dental and Craniofacial Research | DE028599 | Axel Visel |
| National Natural Science Foundation of China | 81771057 | Huan Liu |
| National Natural Science Foundation of China | 81400477 | Huan Liu |
| Natural Science Foundation of Hubei Province | 2017CFB515 | Huan Liu |

The funders had no role in study design, data collection and interpretation, or the decision to submit the work for publication.

## Author contributions

Huan Liu, Conceptualization, Resources, Data curation, Software, Formal analysis, Supervision, Funding acquisition, Validation, Investigation, Visualization, Methodology, Project administration, Writing - drafting, reviewing, editing; Kaylia Duncan, Data curation, Formal analysis, Validation, Investigation, Visualization, Methodology; Annika Helverson, Data curation, Validation, Visualization; Priyanka Kumari, Resources, Data curation, Investigation; Camille Mumm, Yao Xiao, Jenna Colavincenzo Carlson, Elizabeth Leslie, Data curation, Investigation; Fabrice Darbellay, Conceptualization, Data curation, Visualization, Methodology, Writing - review and editing; Axel Visel, Conceptualization, Data curation, Funding acquisition, Investigation, Methodology, Writing - review and editing; Patrick Breheny, Data curation; Albert J Erives, Data curation, Writing - review and editing; Robert A Cornell, Conceptualization, Funding acquisition, Investigation, Project administration, Writing - drafting, reviewing, editing

## Author ORCIDs

Huan Liu ⓘD https://orcid.org/0000-0002-9947-6687
Jenna Colavincenzo Carlson ⓘD http://orcid.org/0000-0001-5483-0833
Axel Visel ⓘD http://orcid.org/0000-0002-4130-7784
Patrick Breheny ⓘD http://orcid.org/0000-0002-0650-1119
Albert J Erives ⓘD http://orcid.org/0000-0001-7107-5518
Robert A Cornell ⓘD https://orcid.org/0000-0003-4207-9100

## Ethics

Animal experimentation: *D. rerio* were maintained in the University of Iowa Animal Care Facility according to a standard protocol (protocol no. 6011616). All mouse experiments were performed in accordance with approval of the Institutional Animal Care and Use Committees at the School and Hospital of Stomatology of Wuhan University (protocol no. 00271454). Mouse experiments for LacZ reporter transgenic animal work performed at the Lawrence Berkeley National Laboratory (LBNL) were reviewed and approved by the LBNL Animal Welfare and Research Committee.

## Decision letter and Author response
Decision letter https://doi.org/10.7554/eLife.51325.sa1
Author response https://doi.org/10.7554/eLife.51325.sa2

# Additional files
## Supplementary files
• Supplementary file 1. Coordinates of ATAC-seq and ChIP-seq peaks identified in this study. (a) Summary of peak numbers for all ATAC-seq and H3K27Ac ChIP-seq generated in this study (b) Coordinates of GFP-positive NFRs flanked by H3K27Ac$^{High}$ (zGPAEs) (c) Coordinates of GFP-positive

NFRs flanked low in H3K27Ac signals (d) Coordinates of GFP-negative NFRs flanked by H3K27Ac$^{High}$ (GNAEs) (e). Coordinates of GFP-negative NFRs flanked low in H3K27Ac signals (f) Coordinates of fish zGPAEs training set (zv9) (g) Coordinates of mouse palate mesenchyme enriched NFR (h) Coordinates of mouse palate epithelium enriched NFR (i) Coordinates of mouse palate epithelium specific active enhancers (j) Coordinates of HIOEC-specific NFRs (k) Coordinates of HIOEC-specific active NFRs (flanked or overlapped with H3K27Ac ChIP-seq in HIOEC)

• Supplementary file 2. Zebrafish *ppl* and human *PPL* enhancer alignments using ClustalO. (a) Alignments summary for enhancer homology test between *ppl-10* and *PPL-8.3*. (b) Alignments details for enhancer homology test between *ppl-10* and *PPL-8.3*. All alignments were conducted using the CLUSTALW algorithm with default parameters via the Clustal Omega server (https://www.ebi.ac.uk/Tools/msa/clustalo/). Alignments were then annotated to highlight identical blocks of length 5 to 6 bp long (cyan) or longer (yellow). See Materials and methods for further details on the choice of enhancer fragments used in these alignments.

• Supplementary file 3. deltaSVM score and JASPAR predicted TF binding changes in the *KRT18* locus. (a) List of OFC-associated SNPs near *KRT18* locus (b) deltaSVM scores for 14 OFC-associated SNPs near KRT18 locus and 1000 random SNPs using classifiers trained by zGPAEs (c) deltaSVM scores for 14 OFC-associated SNPs near KRT18 locus and 1000 random SNPs using classifiers trained by mPEAEs (d) deltaSVM scores for 14 OFC-associated SNPs near KRT18 locus and 1000 random SNPs using classifiers trained by hOEAEs (e) deltaSVM scores for 14 OFC-associated SNPs near KRT18 locus and 1000 random SNPs using classifiers trained by mPMAEs (f) Effects of different alleles of SNP1 and SNP2 on transcription factor binding sites, predicted by JASPAR

• Transparent reporting form

### Data availability

Raw and processed sequencing data were deposited in GEO repository (GSE140241, GSE139945 and GSE139809). Custom scripts and piplines we deployed for sequencing data analysis and visualization are available at https://github.com/Badgerliu/periderm_ATACSeq (copy archived at https://github.com/elifesciences-publications/periderm_ATACSeq). All data generated or analysed during this study are included in the manuscript and supporting files. Source data files have been provided for figures.

The following datasets were generated:

| Author(s) | Year | Dataset title | Dataset URL | Database and Identifier |
|---|---|---|---|---|
| Liu H, Cornell RA | 2020 | Zebrafish periderm at 4-somite stage | https://www.ncbi.nlm.nih.gov/geo/query/acc.cgi?acc=GSE140241 | NCBI Gene Expression Omnibus, GSE140241 |
| Liu H, Cornell RA | 2019 | ATAC-seq profile of mouse palatal epithelium at E14.5 | https://www.ncbi.nlm.nih.gov/geo/query/acc.cgi?acc=GSE139945 | NCBI Gene Expression Omnibus, GSE139945 |
| Liu H, Cornell RA | 2019 | Human oral epithelial cell line HIOEC | https://www.ncbi.nlm.nih.gov/geo/query/acc.cgi?acc=GSE139809 | NCBI Gene Expression Omnibus, GSE139809 |

The following previously published datasets were used:

| Author(s) | Year | Dataset title | Dataset URL | Database and Identifier |
|---|---|---|---|---|
| Bogdanović O, Fernandez-Miñan A, Tena JJ, de la Calle-Mustienes E, Hidalgo C, van Heeringen SJ, Veenstra GJ, Gómez-Skarmeta JL | 2012 | Dynamics of enhancer chromatin signatures mark the transition from pluripotency to cell specification during embryogenesis | https://www.ncbi.nlm.nih.gov/geo/query/acc.cgi?acc=GSE32483 | NCBI Gene Expression Omnibus, GSE32483 |
| ENCODE Pilot Project Research Con- | 2016 | ChIP-seq from embryonic facial prominence (ENCSR481SGM) | https://www.ncbi.nlm.nih.gov/geo/query/acc. | NCBI Gene Expression Omnibus, |

| | | | | cgi?acc=GSE82727 | GSE82727 |
|---|---|---|---|---|---|
| sortium | | | | | |
| NIH Roadmap Epigenomics Mapping Consortium | 2015 | 127-reference epigenome/25-state Imputation Based Chromatin State Model | | http://egg2.wustl.edu/roadmap/data/byFileType/chromhmmSegmentations/ChmmModels/imputed12marks/jointModel/final/ | NIH Roadmap Epigenomics FTP, jointModel/final |

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
