## [Decision Letter]

**Acceptance summary:**

In this interesting paper, the authors identify a set of zebrafish periderm enhancer candidates and use the same methods to identify enhancer candidates in mouse palate epithelium and human oral epithelium cell lines, and then trained a machine learning program (gkm-SVM) on these data sets to identify likely OFC-associated SNPs near the *KRT18* gene, which they functionally tested in reporter assays. The results reveal many important periderm enhancer as well as useful methods for enhance validation.

**Decision letter after peer review:**

Thank you for submitting your article "Analysis of zebrafish periderm enhancers facilitates the identification of a regulatory variant near human *KRT18*" for consideration by *eLife*. Your article has been reviewed by two peer reviewers, and the evaluation has been overseen by Marianne Bronner as the Senior and Reviewing Editor. The following individual involved in review of your submission has agreed to reveal their identity: Alice Goodwin (Reviewer #3).

The reviewers have discussed the reviews with one another and the Reviewing Editor has drafted this decision to help you prepare a revised submission.

Essential revisions:

The reviewers are overall positive but I ask you to discuss the points raised in the individual reviews below, If you are unable to perform the experiments proposed in point 1 of reviewer 2, please discuss whether the snp is expected to alter the binding of a transcription factors. As to the reviewer's second major point, please respond to the point raised.

Reviewer #2:

The data is extensive and the authors do a nice job of integrating their data, including with other data sources. The work on defining periderm enhancers is impressive and very useful for the field. Certainly, parts of this paper confirm previous findings, but it also points to potential new regulators of periderm. The paper does identify a new disease-related SNP in a periderm-related disease. Overall, the paper makes a significant enough advancement.

1) The overall logic is that the definition of zebrafish periderm enhancers is useful for identifying functional SNPs in human periderm-related disease. This is possible despite the limited enhancer conservation from zebrafish to humans because the key transcription factors and overall regulatory logic is better conserved than overall enhancer sequence and because SNPs in regulatory regions disrupt binding of these key transcription factors. In the paper, the authors identify SNP2 near the keratin 18 gene as functional. The approach would be better supported if there was some analysis of which transcription factor(s) might be binding or not binding to SNP2. Also, the prediction would be that this human regulatory region would mediate periderm expression in zebrafish and that this expression was sensitive to the disease variant of SNP2. Was this tested?

2) Given that the machine learning algorithm trained on zebrafish data had a very high false positive rate for predicting periderm enhancers, it seems surprising to me that using that data then for analysis of human enhancers would be very useful. This is compounded by the fact there is no enhancer data on human periderm, leading the authors to use data from cells and cell lines from diverse sources to validate their approach. My worry here is that with the multiple cell lines, some with limited relevance, the authors can show pretty much what they like to show. The authors try to address this by linking the 0.1% top bin scoring tiles with gene expression from mouse periderm, but while the p-value may make a significance cutoff, this does not seem entirely convincing. Of note, the IRF6 enhancer element selected for study by the authors does not make it into the 0.1% top bin scoring tiles. Because of these concerns and the concerns in #1, the generality of the authors' approach for identifying functional disease SNPs in humans is not entirely convincing.

Reviewer #3:

I believe this work is of interest in that it proposes a high throughput method to identify potential functional OFC-associated SNPs. The focus of this article was training on periderm tissue, however, the methods may also be applied to other tissues involved in craniofacial development.

---

## [Author Response]

Reviewer #2:[…] 1) The overall logic is that the definition of zebrafish periderm enhancers is useful for identifying functional SNPs in human periderm-related disease. This is possible despite the limited enhancer conservation from zebrafish to humans because the key transcription factors and overall regulatory logic is better conserved than overall enhancer sequence and because SNPs in regulatory regions disrupt binding of these key transcription factors. In the paper, the authors identify SNP2 near the keratin 18 gene as functional. The approach would be better supported if there was some analysis of which transcription factor(s) might be binding or not binding to SNP2.

We thank the reviewer for recommending this analysis. The revised manuscript now includes these lines:

“We used JASPAR to assess the transcription factor consensus binding sites in 19 bp windows centered on SNP1 (rs11170342) or SNP2 (rs2070875). […] Interestingly, Snai2 is expressed in mouse palate epithelium, and Snai1/Snai2 double mutants exhibit abnormal migration of periderm at medial edge palate epithelium (Murray, Oram and Gridley, 2007).”

Also, the prediction would be that this human regulatory region would mediate periderm expression in zebrafish and that this expression was sensitive to the disease variant of SNP2. Was this tested?

In response to this request, we conducted transient transgenic reporter tests in zebrafish embryos using elements centered on SNP2, our favored SNP, and SNP1, another element of interest. Unexpectedly, both were negative in the zebrafish embryos up to 4 day post fertilization. As a further test of these elements’ enhancer activity, we collaborated with Axel Visel’s group to have them tested in mouse embryos. Again, they were negative, except in one embryo with 8 copies of the element centered on SNP2, which as expected had reporter activity in periderm.

Because both elements are in chromatin marked as enhancers in human embryos and human cell lines, we favor the model that there is a technical explanation underlying these negative results. Non-exclusive possibilities include:

a) we may have failed to amplify large enough regions;

b) the enhancers may not be compatible with the promoters used in the enhancer constructs;

c) they are active at a developmental stage that we did not monitor;

d) the chromatin context in which they integrated is silenced in periderm or perhaps all skin.

The revised manuscript now includes these lines:

“*KRT18* is expressed in many epithelia other than periderm, including trophectoderm (Foshay and Gallicano, 2009), embryonic surface ectoderm (McGowan and Coulombe, 1998; Tadeu and Horsley, 2013), oral epithelia (Gong, Gong and Shum, 2005), embryonic cornea Gong, Gong and Shum, 2005), gonad (Appert et al., 1998), bladder (Eman et al., 2006), choroid plexus (Diez-Roux et al., 2011) and others. […] While not conclusively demonstrating reproducible reporter activity, this result is consistent with the possibility that SNP2 lies in a sequence that has quantitatively mild periderm enhancer activity and may causatively contribute to the phenotypes observed in patients.”

2) Given that the machine learning algorithm trained on zebrafish data had a very high false positive rate for predicting periderm enhancers, it seems surprising to me that using that data then for analysis of human enhancers would be very useful. This is compounded by the fact there is no enhancer data on human periderm, leading the authors to use data from cells and cell lines from diverse sources to validate their approach. My worry here is that with the multiple cell lines, some with limited relevance, the authors can show pretty much what they like to show.

We agree that false positive rates are an important consideration when developing machine learning models for enhancer prediction. However, we are unsure why the reviewer describes our rates as "very high,” because we would describe them as quite low. As shown in the precision recall curve (Figure 3B), we can identify over 50% of all zebrafish enhancers (recall) at a false positive rate of under 10% (precision = 1 – false positive rate). This is quite good; certainly comparable with the false positive rates of other published classifiers.

We disagree that we “can show pretty much what (we) like to show,” but we acknowledge that there is no ground truth upon which to compare the effectiveness of the classifiers. Perhaps the most convincing experiment to test that a given SNP is functional is to engineer a relevant cell line to be homozygous for the risk or non-risk SNP alleles and then compare expression of the relevant gene or genes (in this case *KRT8* and *KRT18*). We have attempted to carry out this experiment with Cas9-mediated homology directed repair (HDR) but have yet to succeed. Short of this experiment, we think the luciferase experiments in the human oral epithelium cell line are the strongest direct test of a SNP’s effect on enhancer activity. In addition, while attempting HDR we generated cells clones homozygous for a small deletion flanking SNP2, and found that *KRT18* expression was lower than normal in such clones, supporting SNP2 being functional.

Describing this experiment, the Results section contains the following text:

“We next tested the prediction that the enhancer in which SNP2 lies regulates expression of *KRT8* and/or *KRT18*. We transfected GMSMK cells with Cas9 ribonucleotide protein (RNP) and, in experimental cells, with two gRNAs targeting sites separated by 109 bp and on either side of SNP2, or, in control cells, with a non-targeting gRNA. […] These results support the notion that the region containing SNP2 is an enhancer driving expression of *KRT8* and *KRT18* in human oral epithelial cells.”

The authors try to address this by linking the 0.1% top bin scoring tiles with gene expression from mouse periderm, but while the p-value may make a significance cutoff, this does not seem entirely convincing. Of note, the IRF6 enhancer element selected for study by the authors does not make it into the 0.1% top bin scoring tiles. Because of these concerns and the concerns in #1, the generality of the authors' approach for identifying functional disease SNPs in humans is not entirely convincing.

We made a stable zebrafish reporter line of the version of IRF6 enhancer that was shown earlier to be active in mouse oral periderm to test our prediction that an element with such activity in mammals would be active in zebrafish surface periderm. This element contains a tile in the 1.0-1.5% of tiles. This score puts it within the range of training set elements. Of note, elsewhere in the larger region marked as an enhancer there is indeed an element in the 0.2% top bin scoring tile.